# DATA SELECTION VIA OPTIMAL CONTROL FOR LANGUAGE MODELS

**Yuxian Gu**[1,2]*,  **Li Dong**[2],  **Hongning Wang**[1]  **Yaru Hao**[2],  **Qingxiu Dong**[3]*,
**Furu Wei**[2],  **Minlie Huang**[1]†
[1]The CoAI Group, Tsinghua University    [2]Microsoft Research    [3]Peking University

## ABSTRACT

This work investigates the selection of high-quality pre-training data from massive corpora to enhance LMs' capabilities for downstream usage. We formulate data selection as a generalized Optimal Control problem, which can be solved theoretically by Pontryagin's Maximum Principle (PMP), yielding a set of necessary conditions that characterize the relationship between optimal data selection and LM training dynamics. Based on these theoretical results, we introduce **P**MP-based **D**ata **S**election (**PDS**), a framework that approximates optimal data selection by solving the PMP conditions. In our experiments, we adopt PDS to select data from CommonCrawl and show that the PDS-selected corpus accelerates the learning of LMs and constantly boosts their performance on a wide range of downstream tasks across various model sizes. Moreover, the benefits of PDS extend to ~400B models trained on ~10T tokens, as evidenced by the extrapolation of the test loss curves according to the Scaling Laws. PDS also improves data utilization when the pre-training data is limited, by reducing the data demand by 1.8 times, which helps mitigate the quick exhaustion of available web-crawled corpora. Our code, model, and data can be found at https://github.com/microsoft/LMOps/tree/main/data_selection.

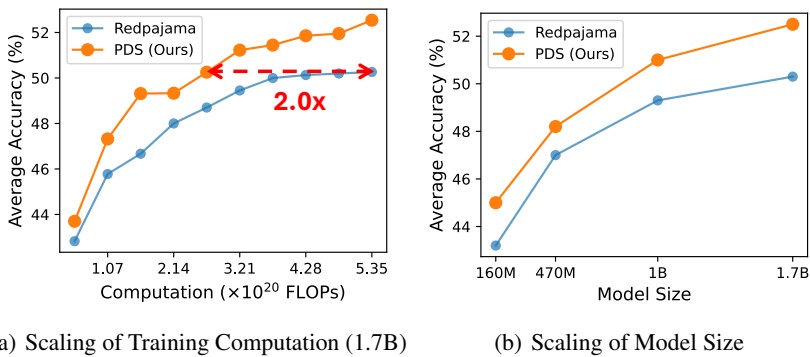

(a) Scaling of Training Computation (1.7B)          (b) Scaling of Model Size

Figure 1: Scaling curves of average accuracy on 9 widely-used downstream tasks with respect to computation (a) and model sizes (b). We select pre-training corpora from the CommonCrawl and pre-train LMs on the selected data. PDS is compared with the Redpajama data cleaning pipeline (Together, 2023). The computation curves are calculated based on the training of a 1.7B LM.

## 1 INTRODUCTION

With the thriving of language models (LMs; Han et al., 2021; Bommasani et al., 2021), the role of **data selection** for pre-training becomes increasingly important, which aims at identifying valuable pre-training instances to accelerate model learning or improve downstream performance (Albalak et al., 2024). This focus enables researchers to explore the limit of LMs in the face of increasing training data demands (Brown et al., 2020; OpenAI, 2023; Team et al., 2023). It also helps to reduce the computational costs during pre-training (Sorscher et al., 2022), and addresses the potential

---

*Contribution during an internship at Microsoft Research. ⟨guyx21@mails.tsinghua.edu.cn⟩
†Corresponding author.

limitations caused by available Internet data (Villalobos et al., 2022; Muennighoff et al., 2023). Without doubt, pre-training data selection is highly valuable for both research and industry sectors.

Unlike previous works relying primarily on manually crafted heuristics (Tirumala et al., 2023; Xie et al., 2023), we connect data selection with classical Optimal Control theory (Lewis et al., 2012), where control variables in a dynamic system are optimized to achieve desired objectives. This mathematical formulation allows fine-grained white-box analysis of how the control variables drive a dynamic system from one state to another. In particular, by treating data selection as the control variables (i.e., whether a data point is included in pre-training), the LM pre-training process as the dynamic system, and the LM's downstream performance as the objective, we leverage Pontryagin's Maximum Principle (PMP; Pontryagin, 2018) to derive the necessary conditions for optimal data selection in theory. These results offer a rigorous, theory-driven alternative to the ad-hoc trial-and-error practices that currently dominate LM pre-training.

Based on our theoretical results, we introduce **PMP**-based **D**ata **S**election (**PDS**), a framework that selects high-quality pre-training data at scale, by solving the equation system induced by the PMP conditions. Balancing effectiveness and efficiency, PDS first solves the equation system for the optimal data selection on a proxy dataset (e.g., 0.2B tokens), assigning a quality score to each instance based on its impact on downstream tasks. After that, a data scorer (e.g., 125M parameters) is trained to predict the quality scores and then infers scores on the target corpus (e.g., 50B tokens). Finally, the predicted scores guide data selection for pre-training LMs with various sizes (e.g., 1.7B parameters).

Unlike previous pre-training data selection methods based on deduplication (Tirumala et al., 2023; Abbas et al., 2023), pattern matching (Xie et al., 2023), or single checkpoint performance (Engstrom et al., 2024), which are agnostic to the pre-training process of LMs, PDS exploits the highly dynamic nature of LM pre-training through the theoretical optimal control perspective. On the other hand, compared to methods that incorporate signals from the LM training process online (Yu et al., 2024; Wang et al., 2024), PDS operates offline, before the training begins, which avoids additional training-time computation overhead and allows for training LMs with arbitrary configurations while performing PDS only once. Furthermore, PDS only filters the training corpus, leaving highly optimized pre-training pipelines largely intact. Most importantly, PDS enjoys a strong theoretical basis, opening up the black box of understanding of individual data point impact on LM pre-training.

In our experiments, we select data from the CommonCrawl with PDS using LIMA (Zhou et al., 2024) as the guide for downstream performance and pre-train LMs with 160M, 470M, 1B, and 1.7B parameters from scratch. Then, we test the LM's zero-shot performance on tasks other than LIMA to examine the generalization of PDS. We observe around 2 times speed-up in pre-training on the 1.7B LM and constant improvement in downstream tasks and language modeling performance across all model sizes compared to state-of-the-art baselines. Extrapolating these results using the Scaling Law (Kaplan et al., 2020; Hoffmann et al., 2022), we show that the benefits remain consistent for ~400B LMs trained on ~15T tokens. Besides, PDS enhances data utilization in a data-constrained setting, reducing the pre-training data demand by 1.8 times, which is a critical advantage as the LM community is running out of data (Villalobos et al., 2022). We also conduct extensive analysis and ablation studies on the key factors of PDS to facilitate further research on data selection.

## 2 METHOD

### 2.1 PROBLEM FORMULATION

We study an LM parameterized with $\boldsymbol{\theta} \in \mathbb{R}^N$, pre-trained from scratch on a dataset $\mathcal{D} = \{x_n\}_{n=1}^{|\mathcal{D}|}$, over $T$ training steps. Data selection (Albalak et al., 2024) aims at identifying a subset $\mathcal{D}'$ from $\mathcal{D}$, such that LMs trained on $\mathcal{D}'$ achieve better performance, measured by a lower downstream loss $J(\boldsymbol{\theta})$.

The pre-training process renders $J(\boldsymbol{\theta})$ as a function of $\mathcal{D}'$, which can be fully characterized by a data quality score vector $\boldsymbol{\gamma} = \left[\gamma_1, \gamma_2, \cdots, \gamma_{|\mathcal{D}|}\right]^\top$ in a $|\mathcal{D}|$-dimensional simplex $U$, where $U = \left\{ \left[\gamma_1, \gamma_2, \cdots, \gamma_{|\mathcal{D}|}\right]^\top \Big| \sum_{n=1}^{|\mathcal{D}|} \gamma_n = 1 \text{ and } \gamma_n \geq 0 \text{ for } 1 \leq n \leq |\mathcal{D}| \right\}$ [1]. A higher quality score in $\boldsymbol{\gamma}$ indicates the corresponding instance is more helpful to reduce $J(\boldsymbol{\theta})$, and thus the LM should learn

---

[1]Only the relative data quality is meaningful for data selection. Therefore, we ensure the sum of the quality scores to 1 to avoid the impact of their individual scales.

more from the instance. Here, we focus on the independent importance of each example and provide a discussion on the dependence between each data point in Appendix E. This results in the following general pre-training loss, defined as the weighted sum of the per-instance loss by $\boldsymbol{\gamma}$:

$$L(\boldsymbol{\theta}, \boldsymbol{\gamma}) = \sum_{n=1}^{|\mathcal{D}|} \gamma_n l(x_n, \boldsymbol{\theta}), \tag{1}$$

where $l(x_n, \boldsymbol{\theta}) = -\log p_{\boldsymbol{\theta}}(x_n)$. The goal of data selection is thus to find $\boldsymbol{\gamma}$ that reduces the downstream loss $J(\boldsymbol{\theta})$, and then select instances with the highest scores in $\boldsymbol{\gamma}$. For simplicity, we assume that the LM is trained using Gradient Decent (GD) for $0 \le t < T$, with the derivation under the Adam optimizer (Kingma & Ba, 2015) provided in Appendix C:

$$\boldsymbol{\theta}_{t+1} = \boldsymbol{\theta}_t - \eta \nabla L(\boldsymbol{\theta}_t, \boldsymbol{\gamma}), \tag{2}$$

where $\boldsymbol{\theta}_t$ represents the model parameters at the time step $t$ during GD and $\eta$ is the learning rate.

**Optimization Problem.** Motivated by the literature on learned optimizers (Metz et al., 2020), we optimize $\boldsymbol{\gamma}$ by minimizing the area under the curve (AUC; Cortes & Mohri, 2003) of $J(\boldsymbol{\theta}_t)$, which is approximated by the cumulative sum of $J(\boldsymbol{\theta}_t)$ over the pre-training process:

$$\min_{\boldsymbol{\gamma}} \ \sum_{t=1}^{T} J(\boldsymbol{\theta}_t),$$
$$\text{s.t. } \boldsymbol{\theta}_{t+1} = \boldsymbol{\theta}_t - \eta \nabla L(\boldsymbol{\theta}_t, \boldsymbol{\gamma}), \ \ \boldsymbol{\gamma} \in U. \tag{3}$$

Intuitively, a lower AUC corresponds to faster convergence of the loss and improved final downstream performance. Unlike evaluating $J(\boldsymbol{\theta}_t)$ at individual time steps, the AUC captures the overall LM training dynamics. As shown in Appendix A, minimizing the AUC essentially enhances the constants in the LM's Scaling Laws (Kaplan et al., 2020), leading to substantial improvements in LM learning.

## 2.2 DATA SELECTION AS OPTIMAL CONTROL

We recognize the optimization problem in Eq. (3) is analogous to a discrete optimal control problem (Lewis et al., 2012), where $J(\cdot)$ is the cost function, the model parameters $\boldsymbol{\theta}$ are the state variables evolving according to Eq. (2), and the data quality scores $\boldsymbol{\gamma}$ are the control variables to be optimized within $U$. This perspective makes it convenient for theoretically solving the data selection problem.

**Theoretically Optimal Solution for Data Selection.** Optimal control problems can be solved by a powerful tool known as **Pontryagin's Maximum Principle** (PMP; Pontryagin, 2018), which provides a set of necessary conditions for the optimal control variables and their corresponding state variables (See Appendix B for its formal expression). However, standard PMP conditions allow the optimal control to vary over time, whereas in Eq. (3), the control variables $\boldsymbol{\gamma}$ are constrained to be **time-invariant** due to the offline nature of data selection in our setting. This typically makes the optimization problem more challenging due to the shrinking of feasible region. In the following, we present the PMP conditions for data selection under this constraint:

**Theorem 2.1** (PMP Conditions for Data Selection). *Let $\boldsymbol{\gamma}^*$ solve the problem in Eq. (3), and $\boldsymbol{\theta}_t^*$ denote the LM parameters trained with $\boldsymbol{\gamma}^*$. For $0 \le t < T$, there exists a vector $\boldsymbol{\lambda}_t^* \in \mathbb{R}^N$ such that*

$$\boldsymbol{\theta}_{t+1}^* = \boldsymbol{\theta}_t^* - \eta \nabla L(\boldsymbol{\theta}_t^*, \boldsymbol{\gamma}^*), \ \ \boldsymbol{\theta}_0^* = \boldsymbol{\theta}_0, \tag{4}$$

$$\boldsymbol{\lambda}_t^* = \boldsymbol{\lambda}_{t+1}^* + \nabla J(\boldsymbol{\theta}_t^*) - \eta \nabla^2 L(\boldsymbol{\theta}_t^*, \boldsymbol{\gamma}^*) \boldsymbol{\lambda}_{t+1}^*, \ \ \boldsymbol{\lambda}_T^* = \nabla J(\boldsymbol{\theta}_T^*), \tag{5}$$

$$\boldsymbol{\gamma}^* = \arg\max_{\boldsymbol{\gamma}} \sum_{n=1}^{|\mathcal{D}|} \gamma_n \left[ \sum_{t=0}^{T-1} \boldsymbol{\lambda}_{t+1}^{*\top} \nabla l(x_n, \boldsymbol{\theta}_t^*) \right], \ \ \boldsymbol{\gamma} \in U, \tag{6}$$

*where $\nabla^2 L(\boldsymbol{\theta}_t^*, \boldsymbol{\gamma}^*)$ denotes the Hessian matrix of $L(\boldsymbol{\theta}, \boldsymbol{\gamma}^*)$ with respect to $\boldsymbol{\theta}$ evaluated at $\boldsymbol{\theta} = \boldsymbol{\theta}_t^*$.*

We prove Theorem 2.1 in Appendix B using the standard PMP conditions and the Lagrange multiplier method, and provide an illustration for this theorem in Figure 2. Since standard PMP conditions and the Lagrange multiplier method are necessary conditions for the optimality, Theorem 2.1 is also a *necessary condition* for optimal data selection. By inspecting the PMP conditions for data selection, we can see that Eq. (4) ensures the LM parameters, trained with the optimal data quality scores, continue to evolve via GD. As illustrated in Figure 2 (Left), Eq. (5) defines $\boldsymbol{\lambda}_t^*$, a "**target vector**"

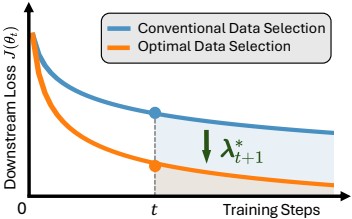 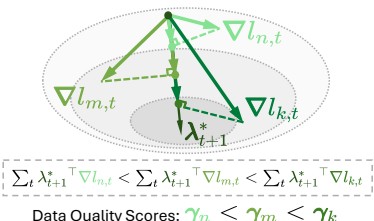

Figure 2: An illustration of Theorem 2.1. **Left**: $\boldsymbol{\lambda}^*_{t+1} \in \mathbb{R}^N$ defines a "target vector" aligning with the optimization direction towards optimal data selection, as in Eq. (5). **Right**: data quality scores are positively correlated with how close the gradient direction of each instance is to the target direction, calculated as the dot-product between $\boldsymbol{\lambda}^*_{t+1}$ and $\nabla l_{i,t} = \nabla l(x_i, \boldsymbol{\theta}^*_t)$ for $i = n, m, k$, as in Eq. (6).

suggesting the ideal gradient direction formed only by high-quality data points. In particular, $\boldsymbol{\lambda}^*_t$ aggregates information about the downstream loss $\nabla J(\boldsymbol{\theta}_t)$ with respect to current training step and the LM's training dynamics $\nabla^2 L(\boldsymbol{\theta}^*_t, \boldsymbol{\gamma}^*)$, from $T$ to $t$. As a result, $\boldsymbol{\lambda}^*_t$ summarizes the dynamics of LM pre-training (i.e., from future to the current). Since $\boldsymbol{\gamma} \in U$, Eq. (6) essentially suggests that $x_n$ with a higher $\sum_t \boldsymbol{\lambda}^*_{t+1}{}^\top \nabla l(x_n, \boldsymbol{\theta}^*_t)$ value should obtain a larger score in $\boldsymbol{\gamma}^*$, as shown in Figure 2 (Right). This indicates that the instances whose gradients align closely with the target vector $\boldsymbol{\lambda}^*_t$ should be selected. Note that the PMP conditions for data selection form a complete equation system, where $\boldsymbol{\theta}^*_t$, $\boldsymbol{\lambda}^*_t$, and $\boldsymbol{\gamma}^*$ are the solutions. In principle, by solving Eqs. (4)-(6) simultaneously, we can derive the optimal data quality scores, forming the foundation of our data selection framework PDS.

## 2.3 PDS: DATA SELECTION BASED ON PMP

PDS selects pre-training data by solving the PMP conditions defined in Eqs. (4)-(6), and consists of three key components, as illustrated in Figure 3. To balance effectiveness and efficiency, PDS first uniformly samples a proxy dataset $\mathcal{D}^{\mathrm{prx}}$ from the pre-training corpus $\mathcal{D}$. Algorithm 1, derived from the PMP conditions, is then applied to $\mathcal{D}^{\mathrm{prx}}$ to compute data quality scores for each instance (Section 2.3.1). Then, a data scorer, typically a small LM, is fine-tuned to predict the quality scores based on the instances in $\mathcal{D}^{\mathrm{prx}}$. The learnt data scorer is subsequently applied to infer quality scores on the entire pre-training corpus $\mathcal{D}$ (Section 2.3.2). Finally, the instances with large scores are selected to form a high-quality corpus $\mathcal{D}'$, which is used to pre-train LMs with any size (Section 2.3.3).

### 2.3.1 DATA QUALITY SCORES FROM PMP

Algorithm 1 solves the PMP conditions for data selection iteratively and returns the data quality scores $\boldsymbol{\gamma}^*$.

**Overview:** Algorithm 1 solves a bi-level optimization problem, consisting of an outer loop to compute $\boldsymbol{\gamma}^*$ and two inner loops to compute $\boldsymbol{\theta}^*_t$ and $\boldsymbol{\lambda}^*_t$ based on the current $\boldsymbol{\gamma}^*$. $\boldsymbol{\gamma}^*$ is first uniformly initialized and then updated for $T_o$ epochs, based on $\boldsymbol{\theta}^*_t$ and $\boldsymbol{\lambda}^*_t$ obtained in the outer iterations.

**Inner loops:** In each iteration of the outer loop, we run the *forward inner loop* to compute $\boldsymbol{\theta}^*_t$ according to Eq. (4) from $t = 0$ to $t = T - 1$, equivalent to training the LM with GD using the current data quality scores to re-weight the per-instance losses. After that, $\boldsymbol{\lambda}^*_t$ is computed from $t = T - 1$ to $t = 0$ with Eq. (5) in the *reverse inner loop*, based on $\boldsymbol{\theta}^*_t$ obtained from the forward inner loop.

---

**Algorithm 1** PMP-Solver

**Input:** LM learning rate $\eta$. Outer loop learning rate $\alpha$. Outer loop epochs $T_o$. Training data before selection $\mathcal{D}$. Downstream loss $J(\boldsymbol{\theta})$. Training steps $T$. $\mathrm{Proj}[\cdot]$ that projects a point in $\mathbb{R}^{|\mathcal{D}|}$ to $U$. Model initialization $\boldsymbol{\theta}_0$.

**Output:** Data quality scores $\boldsymbol{\gamma}^*$.

$\boldsymbol{\gamma} = \left[\gamma_1, \gamma_2, \cdots, \gamma_{|\mathcal{D}|}\right] \leftarrow \left[\frac{1}{|\mathcal{D}|}, \frac{1}{|\mathcal{D}|}, \cdots, \frac{1}{|\mathcal{D}|}\right];$

**repeat** $T_o$ **times**     ▷ Outer loop
  **for** $t = 0, 1, \cdots, T - 1$ **do**     ▷ Forward inner loop
    $\boldsymbol{\theta}_{t+1} \leftarrow \boldsymbol{\theta}_t - \eta \nabla L(\boldsymbol{\theta}_t, \boldsymbol{\gamma})$     ▷ Eq. (4)
  **end for**
  $\boldsymbol{\lambda}_T \leftarrow \nabla J(\boldsymbol{\theta}_T)$
  **for** $t = T - 1, T - 2, \cdots, 1$ **do**     ▷ Reverse inner loop
    $\boldsymbol{\lambda}_t \leftarrow \boldsymbol{\lambda}_{t+1} + \nabla J(\boldsymbol{\theta}_t) - \eta \nabla^2 L(\boldsymbol{\theta}_t, \boldsymbol{\gamma})\boldsymbol{\lambda}_{t+1}$     ▷ Eq. (5)
  **end for**
  **for** $n = 1, 2, \cdots, |\mathcal{D}|$ **do**
    $\gamma_n \leftarrow \gamma_n + \alpha \sum_{t=0}^{T-1} \boldsymbol{\lambda}_{t+1}^\top \nabla l(x_n, \boldsymbol{\theta}_t)$     ▷ Eq. (6)
  **end for**
  $\boldsymbol{\gamma} \leftarrow \mathrm{Proj}[\boldsymbol{\gamma}]$
**end** and **return** $\boldsymbol{\gamma}$

---

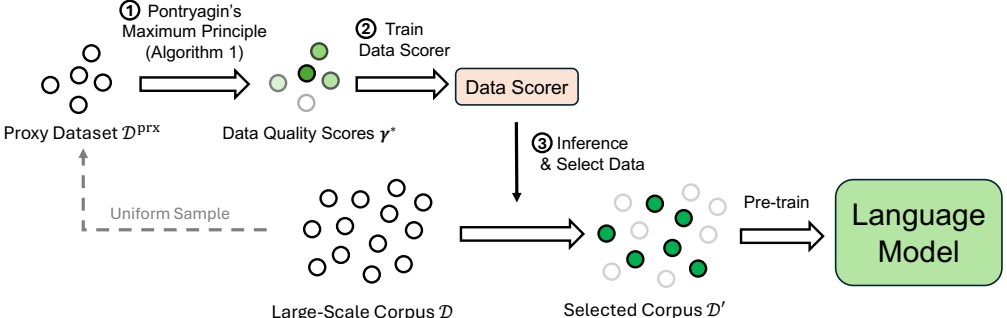

Figure 3: The PDS framework. We compute data quality scores $\boldsymbol{\gamma}^*$ on a proxy dataset $\mathcal{D}^{\mathrm{prx}}$ using Algorithm 1, which is derived from the Pontryagin's Maximum Principle (Pontryagin, 2018) (Section 2.3.1). After that, the data scorer learns to predict quality scores from instances, which then infers scores for a large corpus $\mathcal{D}$ (Section 2.3.2). Finally, a high-quality corpus $\mathcal{D}'$ is selected based on the inferred scores to pre-train an LM (Section 2.3.3).

**Update of $\boldsymbol{\gamma}^*$:** $\boldsymbol{\gamma}^*$ is supposed to be updated according to Eq. (6) with $\boldsymbol{\theta}_t^*$ and $\boldsymbol{\lambda}_t^*$ computed in the inner loops. Eq. (6) indicates that the new $\boldsymbol{\gamma}^*$ should be set as a one-hot vector, where the element corresponding to the highest $\sum_{t=0}^{T-1} \boldsymbol{\lambda}_{t+1}^* {}^\top \nabla l(x_n, \boldsymbol{\theta}_t^*)$ value is set to 1 and the others are set to 0. However, this "hard" update is unstable, as it causes training the LM with only one example in the upcoming outer loop iteration [2]. Therefore, Algorithm 1 adopts a "soft" update, which increases $\boldsymbol{\gamma}^*$ by a value proportional to $\sum_{t=0}^{T-1} \boldsymbol{\lambda}_{t+1}^\top \nabla l(x_n, \boldsymbol{\theta}_t)$ and projects the updated $\boldsymbol{\gamma}^*$ back into $U$.

**Efficient Implementation.** Running Algorithm 1 on $\mathcal{D}^{\mathrm{prx}}$ based on the learning of a large LM remains computationally intensive, as the inner loops involve training the LM for all $T$ steps with GD and computing the Hessian matrix. Therefore, we limit the outer loop to just one epoch and employ stochastic gradient descent (SGD) with a small batch size in the inner loops, which is based on a small proxy LM with $N^{\mathrm{prx}}$ parameters ($N^{\mathrm{prx}} \ll N$) to be trained for $T^{\mathrm{prx}}$ steps ($T^{\mathrm{prx}} \ll T$). To recover any lost long-range training dynamics, we run Algorithm 1 multiple times by setting $\boldsymbol{\theta}_0$ to the checkpoints at different large-scale pre-training stages of the proxy LM and then average the obtained data quality scores on $\mathcal{D}^{\mathrm{prx}}$. Specifically, we first train the proxy LM for $T$ steps and save $M$ checkpoints $\left[\boldsymbol{\theta}_0^{(1)}, \boldsymbol{\theta}_0^{(2)}, \cdots, \boldsymbol{\theta}_0^{(M)}\right]$ in every $\lfloor \frac{T}{M} \rfloor$ steps. Then, the quality scores are given by

$$\boldsymbol{\gamma}^* = \frac{1}{M} \sum_{m=1}^{M} \text{PMP-Solver}\left(\mathcal{D} = \mathcal{D}^{\mathrm{prx}}, T = T^{\mathrm{prx}}, \boldsymbol{\theta}_0 = \boldsymbol{\theta}_0^{(m)}, T_o = 1\right), \tag{7}$$

where PMP-Solver refers to Algorithm 1. We also incorporate several practical optimization techniques to further reduce the computational overhead, as described in Appendix G.1.

### 2.3.2 DATA SCORER

We fine-tune a small LM as the data scorer to fit the data quality scores $\boldsymbol{\gamma}^*$ on $\mathcal{D}^{\mathrm{prx}}$. Specifically, each instance in $\mathcal{D}^{\mathrm{prx}}$ is encoded by averaging the output hidden states of the data scorer. The representation of each instance is then passed through a linear head, outputting a scalar. The linear head and the LM are trained together to fit $\boldsymbol{\gamma}^*$ on $\mathcal{D}^{\mathrm{prx}}$ with the Mean Square Error loss:

$$\boldsymbol{\phi}^*, \boldsymbol{w}^*, b^* = \arg \min_{\boldsymbol{\phi}, \boldsymbol{w}, b} \sum_{n=1}^{|\mathcal{D}^{\mathrm{prx}}|} (\boldsymbol{w}^\top \overline{\boldsymbol{h}}(x_n^{\mathrm{prx}}, \boldsymbol{\phi}) + b - \gamma_n^*)^2, \tag{8}$$

where $\boldsymbol{\phi}$ is the parameters of the data scorer and $\overline{\boldsymbol{h}}(\cdot, \cdot) \in \mathbb{R}^d$ is the average output hidden states of an LM along the sequence length, with $d$ representing the hidden state size. $\boldsymbol{w} \in \mathbb{R}^d, b \in \mathbb{R}$ are the parameters of the linear head. After fine-tuning, we infer the data quality scores of the instances in $\mathcal{D}$ with the data scorer, where the quality score for $x_n$ is given by $\gamma(x_n) = \boldsymbol{w}^*{}^\top \overline{\boldsymbol{h}}(x_n, \boldsymbol{\phi}^*) + b^*$.

---

[2]This does not imply that one single example is optimal for data selection due to the necessity but not sufficiency of Theorem 2.1. See Appendix F for a detailed discussion

### 2.3.3 Data Selection

We use the output scores from the data scorer to estimate the value of the instances in $\mathcal{D}$ to select the final pre-training corpus $\mathcal{D}'$ for the large LM. Given the importance of data diversity in pre-training LMs, we adopt Gumbel-Top-$K$ to introduce randomness into the selection process:

$$\mathcal{D}' = \text{Top-}K \left\{ \gamma(x_n) - \tau \log(-\log(u_n)) \mid x_n \in \mathcal{D}, 1 \le n \le |\mathcal{D}| \right\}, \tag{9}$$

where $u_1, u_2, \cdots, u_{|\mathcal{D}|}$ are independently sampled from $\text{Uniform}(0, 1)$ and $\tau$ is a hyper-parameter to control the strength of the Gumbel noise. The size of the selected data is managed by a data selection ratio $r$, with $K = r|\mathcal{D}|$ in our experiments.

### 2.4 Discussion

**Effectiveness of PDS.** Compared to existing offline data selection methods that curate the pre-training corpus before the LM training starts using pattern information (Xie et al., 2023), deduplication (Tirumala et al., 2023; Abbas et al., 2023), or single-step checkpoints (Engstrom et al., 2024), PDS incorporates long-range training dynamics into data selection, as reflected by the "target vector" $\boldsymbol{\lambda}_t^*$ in Eq. (5). This can be critical for selecting high-quality instances, given the highly dynamic nature of LM pre-training. Although we run Algorithm 1 in a proxy environment and transfer the quality scores to the large-scale setting via the data scorer, many previous works (Xie et al., 2024; Yu et al., 2024) have shown that data quality information is learnable and transferable across model sizes. Different LMs also share many common training dynamics (Tirumala et al., 2022).

**Efficiency and Flexibility of PDS.** Unlike recent online data selection approaches to incorporate LM training dynamics (Yu et al., 2024; Wang et al., 2024), PDS selects the pre-training corpus offline. This allows PDS to be run only once and used for pre-training multiple LMs of any sizes, without incurring additional computational overhead. The FLOPs needed by PDS in a proxy environment are also negligible compared to the demands of large-scale pre-training, as shown in a complexity analysis in Section 3.3. Besides, the offline nature of PDS makes it flexible to be integrated into optimized pre-training pipelines (Chowdhery et al., 2023) by simply replacing the data sources.

## 3 Experiments

### 3.1 Experimental Setup

**Data.** We use the CommonCrawl split from Redpajama (Together, 2023) as $\mathcal{D}$ to exclude the influence of domain weights (Xie et al., 2024). For the downstream loss $J(\boldsymbol{\theta})$, we compute the LM's loss on the training split of LIMA (Zhou et al., 2024), a high-quality dataset consisting of 1,030 diverse instruction-response pairs that cover a wide range of downstream scenarios. The results on more choices of $J(\boldsymbol{\theta})$ are shown in Appendix I.5. Our evaluation is conducted on various downstream datasets other than LIMA to avoid over-fitting.

**Model.** We adopt the same model architecture as Mistral (Jiang et al., 2023) and pre-train LMs with 160M, 470M, 1B, and 1.7B parameters. Model configurations are detailed in Table 6.

**PDS.** To compute the **data quality scores from PMP**, we adopt a 160M proxy LM. $\mathcal{D}^{\text{prx}}$ consists of 160K instances uniformly sampled from $\mathcal{D}$. We first pre-train the proxy LM on $\mathcal{D}$ for 50K steps and then select checkpoints at $[10K, 20K, 30K, 40K, 50K]$ steps. Initialized from these checkpoints, the proxy LM undergoes inner loops with $\eta = 0.008$ over $T^{\text{prx}} = 100$ steps with a mini-batch size of 256. $\boldsymbol{\gamma}^*$ is updated for one outer loop epoch with $\alpha = 1$. For the **data scorer**, we fine-tune a 125M Fairseq-Dense model (Artetxe et al., 2022) along with the linear head, using the objective in Eq. (8). The training details for the data scorer can be found in Appendix G.2. For **Data Selection**, we set $\delta = 0.1$, $r = 0.4$, with further hyper-parameter exploration provided in Appendix I.5.

**Pre-Training.** We pre-train all LMs for 100k steps, using a batch size of 512 and a max input length of 1,024, resulting in roughly 50B tokens. In Section 3.2, we select a 50B-token dataset from a CC corpus containing 125B tokens to assess how different data selection methods improve LM learning given a sufficiently large $\mathcal{D}$. In Section 3.3 (Data-Constrained Setting), we also analyze the effectiveness of PDS when $\mathcal{D}$ is limited in size. See Appendix G.3 for more pre-training details.

| | HS | LAMB | Wino. | OBQA | ARC-e | ARC-c | PIQA | SciQ | BoolQ | Avg. |
|---|---|---|---|---|---|---|---|---|---|---|
| | | | | Model Size = 470M | | | | | | |
| Conventional | 36.7 | 41.4 | 52.4 | **30.4** | 44.8 | 25.2 | 61.0 | 70.6 | 60.4 | 47.0 |
| RHO-Loss | 36.6 | 42.4 | 53.0 | 29.4 | 43.7 | 25.2 | 60.4 | 72.8 | 59.8 | 47.0 |
| DSIR | 36.4 | 42.6 | 51.7 | 29.8 | 46.0 | 24.7 | 61.0 | 72.0 | 55.8 | 46.7 |
| IF-Score | 36.6 | 41.8 | **53.4** | 29.6 | 44.7 | 25.1 | 60.8 | 68.8 | 58.7 | 46.6 |
| PDS | **37.9** | **44.6** | 52.3 | 29.8 | **46.5** | **25.8** | **61.8** | **73.8** | **61.4** | **48.2** |
| | | | | Model Size = 1B | | | | | | |
| Conventional | 39.9 | 47.6 | 52.4 | 30.6 | 49.3 | 26.4 | 63.1 | 73.7 | 60.9 | 49.3 |
| RHO-Loss | 39.8 | 47.0 | 53.0 | 30.8 | 48.0 | 26.4 | 62.9 | 71.1 | **61.0** | 48.9 |
| DSIR | 40.8 | 47.8 | 53.0 | 31.2 | 49.8 | 26.8 | 62.7 | 76.6 | 58.0 | 49.6 |
| IF-Score | 39.4 | 47.0 | 52.6 | 28.6 | 49.4 | 26.4 | 63.5 | 74.0 | 60.5 | 49.0 |
| PDS | **42.1** | **48.8** | **54.0** | **33.4** | **51.3** | **28.0** | **64.1** | **78.5** | 58.7 | **51.0** |

Table 1: Results on the downstream evaluation datasets in OLMo (Groeneveld et al., 2024). We report the accuracy scores and the average scores across the datasets. The best scores of each model size are **boldfaced**. PDS achieves the best performance in most cases compared to the baselines.

**Evaluation.** We evaluate the LMs' 0-shot accuracy on the downstream test datasets used in OLMo (Groeneveld et al., 2024) and their 0-shot performance on MMLU (Hendrycks et al., 2021). We also report the LM's language modeling loss on a subset of DCLM (Li et al., 2024), a high-quality corpus curated with complex pipelines and human heuristics, to verify that models trained on $\mathcal{D}'$ retain diversity and long-tail knowledge coverage. Further evaluation details are in Appendix G.4.

**Baselines.** We compare PDS with conventional pre-training and 3 offline data selection methods:
- Conventional: conventionally pre-training LM on 50B tokens uniformly sampled from $\mathcal{D}$, also refering to "Redpajama" in Figure 1.
- RHO-Loss (Mindermann et al., 2022): selecting data with high reducible losses, as in Eq. (55).
- DSIR (Xie et al., 2023): selecting data with high n-gram overlap with instances in LIMA.
- IF-Score (Koh & Liang, 2017): selecting data with high influence scores, as in Eq. (56).

## 3.2 MAIN RESULTS

**PDS Improves Downstream Performance.** We present the evaluation results of the pre-trained LMs on the OLMo evaluation datasets and MMLU in Table 1 and Table 2, respectively. As shown, PDS outperforms the baselines on most datasets, achieving the best overall performance across models with 470M and 1B parameters. See Appendix I.1 for results on 160M LMs. Figure 1(b) shows the scaling curves of average accuracy on the OLMo evaluation sets with respect to the model sizes, ranging from 160M to 1.7B, demonstrating that the performance improvement remains consistent as the LM scales up. These results indicate that the data quality scores obtained in a proxy environment generalize well to various model sizes and downstream tasks.

| $N$ | Method | 0-shot | PPL |
|---|---|---|---|
| | Conventional | 27.6 | 34.8 |
| | RHO-Loss | 28.4 | 33.0 |
| 470M | DSIR | 28.0 | 34.0 |
| | IF-Score | 28.4 | 31.1 |
| | PDS | **28.9** | **27.1** |
| | Conventional | 29.7 | 26.1 |
| | RHO-Loss | 30.2 | 24.9 |
| 1B | DSIR | 30.0 | 25.3 |
| | IF-Score | 30.7 | 23.6 |
| | PDS | **31.4** | **20.5** |

Table 2: MMLU results. We report the 0-shot accuracy and the perplexity (PPL) on the ground truth answers. The best scores of each model size are **boldfaced**.

**PDS Enhances Language Modeling.** Besides downstream tasks, we show that PDS also enhances language modeling on a high-quality pre-training corpus. Figure 4 shows the test losses on the DCLM subset for conventionally pre-trained LMs and PDS-trained LMs with 160M, 470M, 1B, and 1.7B parameters. We can see that PDS-trained LMs constantly achieve better performance across various model sizes. In Table 3, we extrapolate the test losses using the Scaling Law (Hoffmann et al., 2022), showing that

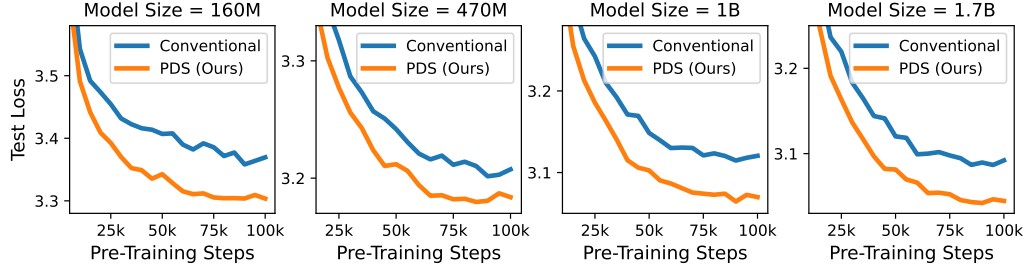

Figure 4: Test losses on the DCLM corpus (Li et al., 2024) for 160M, 470M, 1B and 1.7B LMs.

|  | $N$ | $D$ | Conventional | PDS |
|---|---|---|---|---|
| GPT-3 (Brown et al., 2020) | 175B | 300B | 2.882 | **2.872** |
| Llama (Touvron et al., 2023a) | 6.7B | 1.0T | 2.942 | **2.896** |
| Llama 2 (Touvron et al., 2023b) | 70B | 2.0T | 2.877 | **2.855** |
| Llama 3.1 (Dubey et al., 2024) | 405B | 15T | 2.851 | **2.838** |

Table 3: Test loss extrapolation using the Scaling Law (Hoffmann et al., 2022). We predict the test loss when the LM size $N$ and the trained tokens $D$ meet that of GPT-3 175B, Llama 6.7B, Llama 2 70B, and Llama 3.1 405B. The improvements of PDS remain consistent for these LMs.

the improvements with PDS persist in pre-training recent large LMs, such as GPT-3 (Brown et al., 2020) and Llama family (Touvron et al., 2023a;b; Dubey et al., 2024). Details of the extrapolation are provided in Appendix I.4. The DCLM corpus is curated using a complex pipeline based on human heuristics and is verified to be diverse and comprehensive in its coverage of human knowledge. Algorithm 1 offers a principled alternative to the complex pipeline for curating pre-training corpus.

**PDS Accelerates LM Learning.** In Figure 1(a), we plot the average accuracy scores on the OLMo evaluation datasets with respect to the training FLOPs of the 1.7B model. PDS achieves 2.0 times acceleration in terms of training-time computation compared to conventional pre-training. Similar trends are observed for other model sizes (Figure 8) and DCLM losses (Figure 4).

## 3.3 ANALYSIS

**Data-Constrained Setting.** In Section 3.2, we assume the original pre-training data is unlimited to ensure that the PDS-selected corpus contains 50B tokens, the same as in conventional pre-training. In practice, when the data before selection is limited, the LM has to be trained on the PDS-selected data for multiple epochs. In Figure 5, we restrict $\mathcal{D}$ to contain 50B tokens and apply PDS with selection ratios $r \in [0.125, 0.25, 0.5]$, corresponding to training the LM on the selected data for $[8, 4, 2]$ epochs, respectively. We can see that selecting 1/4 data with PDS and training for 4 epochs achieves the lowest test losses, consistent with the findings of Muennighoff et al. (2023). This suggests that PDS improves data utilization as the high-quality web-crawled corpora run out (Villalobos et al., 2022). We extrapolate the loss curve of conventional pre-training with the Scaling Law suggested by (Muennighoff et al., 2023), and find that it requires another 42B tokens to achieve the performance of PDS ($r = 0.25$, 4 Eps), which means PDS reduces the pre-training data requirement by 1.8 times. See Appendix I.4 for details of this extrapolation.

**Efficient Implementation for Solving Data Quality Scores.** In Section 2.3.1, we introduce efficient implementations to solve the data quality scores by running Algorithm 1 for a single outer epoch, leveraging a small proxy LM trained for a limited number of steps. In Figure 6, we use a simulated setting where $\gamma^*$ can be obtained exactly from Algorithm 1 within an affordable computational budget and compare its performance (measured by $J(\boldsymbol{\theta}_t)$) and overhead (measured in FLOPs) with our efficient implementation. Details of this simulated setting can be found in Appendix G.6. The results show that the efficient implementation significantly reduces computational overhead while preserving most of the performance of the exact solution.

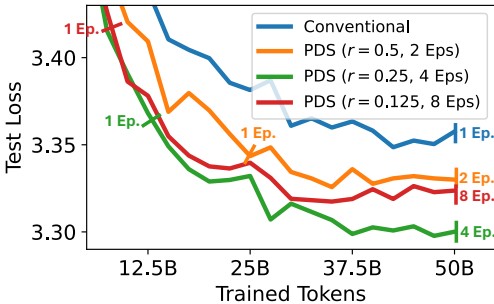
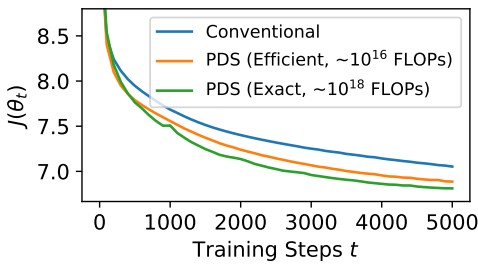

Figure 5: Test losses on DCLM corpus (Li et al., 2024) in the data-constrained setting. We select data with PDS for different selection ratios $r$ and train the model for multiple epochs to reach the same token number budgets.

Figure 6: Comparison between exact and efficient implementation to solve the data quality scores in a simulated setting. The effectiveness is measured by $J(\boldsymbol{\theta}_t)$. The efficient implementation saves computation and preserves most of the performance of the exact solution.

| | | Complexity | FLOPs ($\times 10^{20}$) | Actual Time |
|---|---|---|---|---|
| | Proxy $\gamma$-solver | $O(N^{\text{prx}}D + 4MN^{\text{prx}}D^{\text{prx}})$ | 0.49 | 15.2 Hours |
| PDS | Data Scorer | $O(3N^{\text{score}}D^{\text{prx}} + N^{\text{score}}D)$ | 0.063 | 1.50 Hours |
| | Data Selection | $O(D)$ | 0.0 | 10.2 Minutes |
| Pre-Training | | $O(ND)$ | 5.1 | 144 Hours |

Table 4: Asymptotic complexity, GPU FLOPs, and actually spent time of different PDS steps and 1.7B model pre-training. $N^{\text{prx}}$ and $D^{\text{prx}}$ are the sizes of the proxy LM and proxy data. $N^{\text{score}}$ is the size of the data scorer. We elaborate on the details of how the complexity and FLOPs are computed in Appendix H. PDS costs little computational overhead compared to pre-training large LMs.

**Complexity Analysis.** We compare the computational complexity of PDS with pre-training in Table 4. The overhead of running PDS to select data is only about 1/9 of that of pre-training a 1.7B model. More importantly, since PDS is an offline method, the selected corpus can be used to pre-train any number of LMs without additional computational cost. In addition, the offline nature of PDS allows it to seamlessly integrate into recent highly optimized pre-training pipelines (Chowdhery et al., 2023), requiring only a replacement of the data source without altering the pre-training process.

## 3.4 ABLATION STUDIES ON PMP-SOLVER

**Training Dynamics Information.** Incorporating the LMs' training dynamics into data selection is a key distinction between PDS and other offline data selection approaches. While IF-Score also uses the gradient information of well-trained LMs to estimate data importance, we find that the *long-range* training dynamics in *early training stages* are more valuable. Table 5 shows the results when different types of learning information are considered. PDS (50K) refers to using the LM checkpoint at 50K as $\boldsymbol{\theta}_0^{(m)}$ in Eq. (7). PDS ($T^{\text{prx}} = 1$) means running the inner loops as in Eq. (4) and Eq. (5) for only one step, excluding long-range information. PDS (50K-100K) refers to setting $\boldsymbol{\theta}_0^{(m)}$ to checkpoints at later training stages. Comparing our choice with IF-Score, PDS (50K), and PDS ($T^{\text{prx}} = 1$), we conclude that the long-range training dynamics is more valuable than single-step gradient, although it may be slightly harder for the data scorer to fit. Our choice also outperforms PDS (50K-100K), indicating that the early-stage training dynamics are more beneficial than those from later stages. An explanation is that LMs conduct large-range parameter searching during early training and converge to local optimums in the late stages. Therefore, early-stage information helps the LM choose better local optimums, which can always be achieved by later annealing.

**Proxy Model and Proxy Data.** In Figure 7, we explore how the sizes of the proxy LM and the proxy dataset affect the performance of the data scorer and the pre-trained LM. As the size of the proxy LM increases, the LM's downstream performance increases, but the data scorer's performance

| | Corr. | Acc. |
|---|---|---|
| Conventional | - | 43.2 |
| IF-Score | 0.32 | 43.0 |
| PDS (50K) | 0.51 | 44.0 |
| PDS ($T^{\mathrm{prx}}$=1) | 0.54 | 44.6 |
| PDS (50K-100K) | 0.48 | 43.4 |
| PDS (10K-50K, ours) | 0.52 | 45.0 |

Table 5: Data selection based on different kinds of learning information. We report the LM's zero-shot accuracy on the OLMo evaluation tasks (Acc.) and the Spearman Correlation of the data scorer (Corr.).

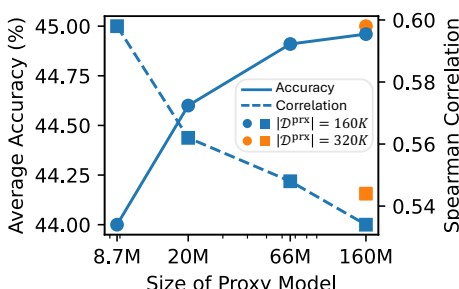

Figure 7: LM performance on the OLMo evaluation tasks (Average Accuracy) and data scorer performance (Spearman Correlation) when different proxy model and proxy data sizes are adopted.

decreases. We notice that when using the 8.7M model, the LM's performance (44.0) is close to that of DSIR (43.8), which selects data based on n-gram matching. This implies that small LMs estimate data quality primarily through shallow patterns that are easy to learn. More complex LM learning information is encoded in the data quality scores computed based on larger proxy LMs, making it harder for the data scorer to fit, but this can be mitigated by increasing the size of $\mathcal{D}^{\mathrm{prx}}$.

## 4 RELATED WORK

**Data Selection for Language Models.** Many works have explored data selection approaches to accelerate LM learning or improve downstream performance (Albalak et al., 2024). Some curate data prior to LM training, which we refer to offline methods, including domain-mixing (Xie et al., 2024; Fan et al., 2023; Gao et al., 2020), data pruning (Marion et al., 2023; Tirumala et al., 2023; Abbas et al., 2023), sample-wise data selection (Xia et al., 2024b; Du et al., 2023; Xie et al., 2023; Gu et al., 2023), and data programming (Ratner et al., 2016; Gu et al., 2022a;b). Other works dynamically select data during LM training by adjusting domain-mixing weights (Xia et al., 2024a; Chen et al., 2024; Albalak et al., 2023) or more fine-grained reweighting strategies (Fan & Jaggi, 2023; Grangier et al., 2023; Gu et al., 2024; Thakkar et al., 2023). This work studies data selection from its general principles, theoretically deriving optimal selection and designing scalable algorithms to implement it.

**Optimal Control in Deep Learning.** The principles of optimal control (Lewis et al., 2012) have been shown to be effective in deep learning (Benning et al., 2019; Liu & Theodorou, 2019), by treating the forward pass of a multi-layer neural network as a control process where the hidden vectors are state parameters and the model parameters are control variables to optimize. With the help of Pontryagin's Maximum Principle (Pontryagin, 2018), some works design optimization algorithms with better convergence rates (Li et al., 2017; Zhang et al., 2019) or broader application scenarios (Li & Hao, 2018), and others provide theoretical foundations of neural networks for better interpretation (Han et al., 2019; Geshkovski & Zuazua, 2022). Unlike these works, we adopt Optimal Control in a novel and "orthogonal" way, by treating the model's learning pass as the control process.

## 5 CONCLUSION

In this work, we investigate selecting pre-training data of LMs from both theoretical and practical perspectives. We first formulate data selection as an Optimal Control problem to derive a set of necessary conditions for optimal data selection using Pontryagin's Maximum Principle (PMP), which establishes theoretical principles for LM pre-training. Then, building on these theoretical results, we introduce PDS, a practical framework that solves the PMP conditions in practice based on long-range LM training dynamics. Extensive experiments show that PDS selects high-quality pre-training corpora that accelerate LM learning and boost LM performance across various model sizes and downstream tasks. We find that PDS also improves data utilization in data-constrained settings, which mitigates the pre-training data exhaustion problem.

## ACKNOWLEDGEMENTS

This work is supported by the National Science Foundation for Distinguished Young Scholars (with No. 62125604) and Tsinghua University Initiative Scientific Research Program.

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

## A  CONNECTION BETWEEN AUC AND SCALING LAW CONSTANTS

We show that the area under the loss curve (AUC) is directly connected to the scaling law (Kaplan et al., 2020) constants, and minimizing AUC essentially improves the scaling properties of LMs. As suggested by existing literature (Kaplan et al., 2020; Hoffmann et al., 2022), the test losses of LMs follow a power-law with respect to the training steps $t$ after a warmup stage:

$$L(t) = \frac{C}{t^c} + L^{\text{irre}}, \ \ t > T_0, \tag{10}$$

where $C$ and $c$ are scaling law constants, $L^{\text{irre}}$ is the irreducible loss related to the noise in the test set, and $T_0$ is the end steps of the warmup stage. $L^{\text{irre}}$ is invariant to optimizing pre-training data selection strategies. Therefore, we care about the reducible loss, whose constants depend on the data quality scores $\boldsymbol{\gamma}$:

$$L^{\text{re}}(t) = L(t) - L^{\text{irre}} = \frac{C(\boldsymbol{\gamma})}{t^{c(\boldsymbol{\gamma})}}, \ \ t > T_0. \tag{11}$$

We then consider the AUC of the reducible loss for sufficiently large $T$:

$$\text{AUC}(\boldsymbol{\gamma}) = \int_{t=T_0}^{T} \frac{C(\boldsymbol{\gamma})}{t^{c(\boldsymbol{\gamma})}} \mathrm{d}t = \frac{C(\boldsymbol{\gamma})}{1 - c(\boldsymbol{\gamma})}(T^{1-c(\boldsymbol{\gamma})} - T_0^{1-c(\boldsymbol{\gamma})}). \tag{12}$$

For $c(\boldsymbol{\gamma}) < 1$, when $T$ is sufficiently large, $\text{AUC}(\boldsymbol{\gamma}) \approx \frac{C(\boldsymbol{\gamma})}{1-c(\boldsymbol{\gamma})}T^{1-c(\boldsymbol{\gamma})}$. Minimizing AUC causes $C(\boldsymbol{\gamma})$ to decrease and $c(\boldsymbol{\gamma})$ to increase[3], improving the scaling properties of LMs. For $c(\boldsymbol{\gamma}) > 1$, $\text{AUC}(\boldsymbol{\gamma}) \approx \frac{C(\boldsymbol{\gamma})}{c(\boldsymbol{\gamma})-1}\frac{1}{T_0^{c(\boldsymbol{\gamma})-1}}$, which also exhibit a trend of decreasing $C(\boldsymbol{\gamma})$ and increasing $c(\boldsymbol{\gamma})$ when AUC is minimized.

## B  PROOF OF THEOREM 2.1

To prove Theorem 2.1, we first describe the standard discrete-time Pontryagin's Maximum Principle (PMP; Pontryagin, 2018) in Optimal Control (Lewis et al., 2012) for *time-variant* control variables:

**Theorem B.1** (PMP). *Consider the following optimization problem in a discrete dynamical system:*

$$\min_{\boldsymbol{\gamma}_t} \sum_{t=0}^{T-1} \mathcal{J}(\boldsymbol{\theta}_t, \boldsymbol{\gamma}_t) + J(\boldsymbol{\theta}_T), \tag{13}$$

$$\textit{s.t. } \boldsymbol{\theta}_{t+1} = f(\boldsymbol{\theta}_t, \boldsymbol{\gamma}_t), \ \ \boldsymbol{\gamma}_t \in U,$$

*where the state variable $\boldsymbol{\theta}_t \in \mathbb{R}^N$, the control variable $\boldsymbol{\gamma}_t \in \mathbb{R}^D$, and $\mathcal{J} : \mathbb{R}^{N \times D} \mapsto \mathbb{R}$, $f : \mathbb{R}^{N \times D} \mapsto \mathbb{R}^N$ are continuous in $\mathbb{R}^{N \times D}$. Let $\boldsymbol{\gamma}_t^*$ be the solution to this problem, and $\boldsymbol{\theta}_t^*$ denote the corresponding state variable. For $0 \le t < T$, there exists a co-state vector $\boldsymbol{\lambda}_t^* \in \mathbb{R}^N$ such that*

$$\boldsymbol{\theta}_{t+1}^* = \nabla_{\boldsymbol{\lambda}} H(\boldsymbol{\theta}_t^*, \boldsymbol{\lambda}_{t+1}^*, \boldsymbol{\gamma}_t^*), \ \ \boldsymbol{\theta}_0^* = \boldsymbol{\theta}_0, \tag{14}$$

$$\boldsymbol{\lambda}_t^* = \nabla_{\boldsymbol{\theta}} H(\boldsymbol{\theta}_t^*, \boldsymbol{\lambda}_{t+1}^*, \boldsymbol{\gamma}_t^*), \ \ \boldsymbol{\lambda}_T^* = \nabla J(\boldsymbol{\theta}_T), \tag{15}$$

$$\boldsymbol{\gamma}_t^* = \arg\min_{\boldsymbol{\gamma}_t} H(\boldsymbol{\theta}_t^*, \boldsymbol{\lambda}_{t+1}^*, \boldsymbol{\gamma}_t), \ \ \boldsymbol{\gamma}_t \in U, \tag{16}$$

*where $H : \mathbb{R}^N \times \mathbb{R}^N \times \mathbb{R}^D \mapsto \mathbb{R}$ is the Hamiltonian function defined by*

$$H(\boldsymbol{\theta}, \boldsymbol{\lambda}, \boldsymbol{\gamma}) = \mathcal{J}(\boldsymbol{\theta}, \boldsymbol{\gamma}) + \boldsymbol{\lambda}^\top f(\boldsymbol{\theta}, \boldsymbol{\gamma}). \tag{17}$$

Proof of Theorem B.1 can be found in various textbooks in Optimal Control (Lewis et al., 2012; Evans, 2005). In our context, we interpret $\boldsymbol{\theta}_t$ as the model parameters, $J(\cdot)$ as the downstream loss function, $U$ as the $|\mathcal{D}|$-dimensional simplex as defined in Section 2.1 and $f$ as the GD operation where the data weights $\boldsymbol{\gamma}_t$ changes with respect to the training steps $t$:

$$\boldsymbol{\theta}_{t+1} = \boldsymbol{\theta}_t - \eta \nabla \sum_{n=1}^{|\mathcal{D}|} \gamma_{n,t} l(x_n, \boldsymbol{\theta}_t) \tag{18}$$

$$= \boldsymbol{\theta}_t - \eta \nabla L(\boldsymbol{\theta}_t, \boldsymbol{\gamma}_t).$$

---

[3]$f(c) = \frac{1}{1-c}T^{1-c}$ is increasing with respect to $c$ when $1 - c < \ln T$, which is easily satisfied for sufficiently large $T$.

In this way, the Hamilton function is

$$H(\boldsymbol{\theta}, \boldsymbol{\lambda}, \boldsymbol{\gamma}) = \mathcal{J}(\boldsymbol{\theta}, \boldsymbol{\gamma}) + \boldsymbol{\lambda}^\top \left[ \boldsymbol{\theta} - \eta \nabla L(\boldsymbol{\theta}, \boldsymbol{\gamma}) \right]. \tag{19}$$

The key challenge to prove Theorem 2.1 is that the control variables are constrained to be invariant of the training step $t$ in data selection, as discussed in Section 2.1 and 2.2, and introducing more constraint usually makes an optimization problem more complex. Formally, the requirement of invariant data weights can be expressed by $T - 1$ equations:

$$\boldsymbol{\gamma}_1 = \boldsymbol{\gamma}_0, \ \boldsymbol{\gamma}_2 = \boldsymbol{\gamma}_0, \cdots, \boldsymbol{\gamma}_{T-1} = \boldsymbol{\gamma}_0. \tag{20}$$

Therefore, the optimization of data selection, as in Eq. (3), is equivalent to the following problem:

$$\begin{aligned}
&\min_{\boldsymbol{\gamma}_t} \sum_{t=1}^{T} J(\boldsymbol{\theta}_t), \\
&\text{s.t. } \boldsymbol{\theta}_{t+1} = \boldsymbol{\theta}_t - \eta \nabla L(\boldsymbol{\theta}_t, \boldsymbol{\gamma}_t), \ \ \boldsymbol{\gamma}_t \in U, \\
&\quad \boldsymbol{\gamma}_0 = \boldsymbol{\gamma}_1 = \cdots = \boldsymbol{\gamma}_{T-1}.
\end{aligned} \tag{21}$$

We adopt the method of Lagrange multipliers to solve Eq. (21) by considering the following optimization problem:

$$\begin{aligned}
&\min_{\boldsymbol{\gamma}_t} \sum_{t=0}^{T-1} J(\boldsymbol{\theta}_t) + \sum_{t=1}^{T-1} \sum_{n=1}^{|\mathcal{D}|} \mu_{n,t} (\gamma_{n,t} - \gamma_{n,0}) + J(\boldsymbol{\theta}_T), \\
&\text{s.t. } \boldsymbol{\theta}_{t+1} = \boldsymbol{\theta}_t - \eta \nabla L(\boldsymbol{\theta}_t, \boldsymbol{\gamma}_t), \ \ \boldsymbol{\gamma}_t \in U,
\end{aligned} \tag{22}$$

where $(\mu_{n,t})_{1 \le n \le D, 0 \le t \le T-1}$ are Lagrange multipliers. Note that we split $J(\boldsymbol{\theta}_T)$ out and add $J(\boldsymbol{\theta}_0)$ to the sum of $J(\boldsymbol{\theta}_t)$, which does not affect the minimization. In this way, when $\mathcal{J}(\boldsymbol{\theta}_t, \boldsymbol{\gamma}_t)$ takes the following form:

$$\mathcal{J}(\boldsymbol{\theta}_t, \boldsymbol{\gamma}_t) = \begin{cases} J(\boldsymbol{\theta}_0) - \displaystyle\sum_{t'=1}^{T-1} \sum_{n=1}^{|\mathcal{D}|} \mu_{n,t'} \gamma_{n,0}, & \text{if } t = 0 \\[4mm] J(\boldsymbol{\theta}_t) + \displaystyle\sum_{n=1}^{|\mathcal{D}|} \mu_{n,t} \gamma_{n,t}, & \text{if } 1 \le t \le T-1 \end{cases}, \tag{23}$$

we can apply Theorem B.1 to Eq. (21). By substituting Eq. (19), Eq. (20), and Eq. (23) into Eq. (14) and Eq. (15), we have:

$$\boldsymbol{\theta}_t^* = \boldsymbol{\theta}_t^* - \eta \nabla L(\boldsymbol{\theta}_t^*, \boldsymbol{\gamma}_0^*), \quad \boldsymbol{\theta}_0^* = \boldsymbol{\theta}_0, \tag{24}$$

$$\boldsymbol{\lambda}_t^* = \boldsymbol{\lambda}_{t+1}^* + \nabla J(\boldsymbol{\theta}_t^*) - \eta \nabla^2 L(\boldsymbol{\theta}_t^*, \boldsymbol{\gamma}_0^*) \boldsymbol{\lambda}_{t+1}^*, \quad \boldsymbol{\lambda}_T^* = \nabla J(\boldsymbol{\theta}_T), \tag{25}$$

which prove Eq. (4) and Eq. (5) in Theorem 2.1 when we set $\boldsymbol{\gamma}_0^* = \boldsymbol{\gamma}^*$. By substituting Eq. (19) and Eq. (23) into Eq. (16), we have

$$\boldsymbol{\gamma}_t^* = \begin{cases} \arg\max_{\boldsymbol{\gamma}_0} \displaystyle\sum_{n=1}^{|\mathcal{D}|} \gamma_{n,0} \left[ {\boldsymbol{\lambda}_1^*}^\top \nabla l(x_n, \boldsymbol{\theta}_0^*) + \sum_{t'=1}^{T-1} \mu_{n,t'} \right], & \text{if } t = 0 \\[4mm] \arg\max_{\boldsymbol{\gamma}_t} \displaystyle\sum_{n=1}^{|\mathcal{D}|} \gamma_{n,t} \left[ {\boldsymbol{\lambda}_{t+1}^*}^\top \nabla l(x_n, \boldsymbol{\theta}_t^*) - \mu_{n,t} \right], & \text{if } 1 \le t \le T-1 \end{cases} \tag{26}$$

Considering the time-invariant constraint in Eq. (20), we set $\boldsymbol{\gamma}_0^* = \boldsymbol{\gamma}_1^* = \cdots = \boldsymbol{\gamma}_{T-1}^* = \boldsymbol{\gamma}^*$ and get

$$\begin{cases} \boldsymbol{\gamma}^* = \arg\max_{\boldsymbol{\gamma}} \displaystyle\sum_{n=1}^{|\mathcal{D}|} \gamma_n \left[ \eta {\boldsymbol{\lambda}_1^*}^\top \nabla l(x_n, \boldsymbol{\theta}_0^*) + \sum_{t'=1}^{T-1} \mu_{n,t'} \right], & \text{if } t = 0 \\[4mm] \boldsymbol{\gamma}^* = \arg\max_{\boldsymbol{\gamma}} \displaystyle\sum_{n=1}^{|\mathcal{D}|} \gamma_n \left[ \eta {\boldsymbol{\lambda}_{t+1}^*}^\top \nabla l(x_n, \boldsymbol{\theta}_t^*) - \mu_{n,t} \right], & \text{if } 1 \le t \le T-1 \end{cases}, \tag{27}$$

which forms a complete equation system containing $T$ equations and $T$ unknown variables ($T-1$ number of $\boldsymbol{\mu}_t = \left[\mu_{1,t}, \mu_{2,t}, \cdots, \mu_{|\mathcal{D}|,t}\right]$ plus one $\boldsymbol{\gamma}^*$), which has the solution:

$$\mu_{n,t} = \eta {\boldsymbol{\lambda}_{t+1}^*}^\top \nabla l(x_n, \boldsymbol{\theta}_t^*) - \eta \frac{S_n}{T}, \ \ 1 \le n \le |\mathcal{D}|, \ \ 0 \le t \le T-1, \tag{28}$$

$$\boldsymbol{\gamma}^* = \arg\max_{\boldsymbol{\gamma}} \sum_{n=1}^{|\mathcal{D}|} \gamma_n \eta \frac{S_n}{T}, \tag{29}$$

where $S_n = \sum_{t=0}^{T-1} {\boldsymbol{\lambda}_{t+1}^*}^\top \nabla l(x_n, \boldsymbol{\theta}_t^*)$. Note that Eq. (29) is equivalent to Eq. (6). So far, Theorem 2.1 is proved by combining Eq. (24), Eq. (25), and Eq. (29).

## C    DERIVATION FOR ADAM

We provide our formulation and derivation for Adam (Kingma & Ba, 2015) in this section. For $0 \le t \le T-1$, the parameter update rules of Adam is given by

$$\begin{aligned}
\boldsymbol{m}_{t+1} &= \beta_1 \boldsymbol{m}_t + (1-\beta_1)\boldsymbol{g}_t, \\
\boldsymbol{v}_{t+1} &= \beta_2 \boldsymbol{v}_t + (1-\beta_2)\boldsymbol{g}_t^2, \\
\widehat{\boldsymbol{m}}_{t+1} &= \boldsymbol{m}_{t+1}/(1-\beta_1^{t+1}), \\
\widehat{\boldsymbol{v}}_{t+1} &= \boldsymbol{v}_{t+1}/(1-\beta_2^{t+1}), \\
\boldsymbol{\theta}_{t+1} &= \boldsymbol{\theta}_t - \eta \widehat{\boldsymbol{m}}_{t+1}/(\sqrt{\widehat{\boldsymbol{v}}_{t+1}} + \epsilon),
\end{aligned} \tag{30}$$

where $\beta_1$, $\beta_2$, $\epsilon$, $\eta$ are hyper-parameters and $\boldsymbol{g}_t = \nabla L(\boldsymbol{\theta}_t, \boldsymbol{\gamma}_t, \mathcal{D})$. We set $\boldsymbol{m}_0 = \mathbf{0}$ and $\boldsymbol{v}_0 = \mathbf{0}$. To formulate the LM training with Adam as an Optimal Control (Lewis et al., 2012) problem, we treat the vector $\boldsymbol{\Theta}_t = [\boldsymbol{\theta}_t, \boldsymbol{m}_t, \boldsymbol{v}_t]^\top \in \mathbb{R}^{3N}$ as the state variable. Let $F$ denote the state-transition function from $\boldsymbol{\Theta}_t$ to $\boldsymbol{\Theta}_{t+1}$:

$$\boldsymbol{\theta}_{t+1} = F(\boldsymbol{\Theta}_t, \boldsymbol{\gamma}), \tag{31}$$

and thus $F$ represents the following relations:

$$\boldsymbol{\theta}_{t+1} = \boldsymbol{\theta}_t - \frac{\eta}{1-\beta_1^{t+1}} \frac{\beta_1 \boldsymbol{m}_t + (1-\beta_1)\boldsymbol{g}_t}{\sqrt{(\beta_2 \boldsymbol{v}_t + (1-\beta_2)\boldsymbol{g}_t^2)/(1-\beta_2^{t+1})} + \epsilon}, \tag{32}$$

$$\boldsymbol{m}_{t+1} = \frac{\beta_1 \boldsymbol{m}_t + (1-\beta_1)\boldsymbol{g}_t}{1-\beta_1^{t+1}}, \tag{33}$$

$$\boldsymbol{v}_{t+1} = \frac{\beta_2 \boldsymbol{v}_t + (1-\beta_2)\boldsymbol{g}_t^2}{1-\beta_2^{t+1}}. \tag{34}$$

Similar to GD, we can still define the Hamiltonian function of the problem and obtain the following theorem with Pontryagin's Maximum Principle (PMP; Pontryagin, 2018):

**Theorem C.1** (PMP Data Selection for Adam). *Let $\boldsymbol{\gamma}^*$ solve the problem in Eq. (3), and $\boldsymbol{\Theta}_t^*$ denote the state variable corresponding to $\boldsymbol{\gamma}^*$. Then, there exists a co-state vector $\boldsymbol{\Lambda}_t^* \in \mathbb{R}^{3N}$ such that*

$$\boldsymbol{\Theta}_{t+1}^* = \nabla_{\boldsymbol{\Lambda}} \mathcal{H}(\boldsymbol{\Theta}_t^*, \boldsymbol{\Lambda}_{t+1}^*, \boldsymbol{\gamma}^*), \ \ \boldsymbol{\Theta}_0^* = [\boldsymbol{\theta}_0, \mathbf{0}, \mathbf{0}]^\top, \tag{35}$$

$$\boldsymbol{\Lambda}_t^* = \nabla_{\boldsymbol{\Theta}} \mathcal{H}(\boldsymbol{\Theta}_t^*, \boldsymbol{\Lambda}_{t+1}^*, \boldsymbol{\gamma}^*), \ \ \boldsymbol{\Lambda}_T^* = [\nabla J(\boldsymbol{\theta}_T), \mathbf{0}, \mathbf{0}]^\top, \tag{36}$$

$$\boldsymbol{\gamma}^* = \arg\min_{\boldsymbol{\gamma}} \sum_{t=0}^{T-1} \mathcal{H}(\boldsymbol{\Theta}_t^*, \boldsymbol{\Lambda}_{t+1}^*, \boldsymbol{\gamma}), \ \ \boldsymbol{\gamma} \in U, \tag{37}$$

*where $\mathcal{H} : \mathbb{R}^{3N} \times \mathbb{R}^{3N} \times \mathbb{R}^D \mapsto \mathbb{R}$ is the Hamiltonian function defined by*

$$\mathcal{H}(\boldsymbol{\Theta}, \boldsymbol{\lambda}, \boldsymbol{\gamma}) = J(\boldsymbol{\theta}) + \boldsymbol{\Lambda}^\top F(\boldsymbol{\Theta}, \boldsymbol{\gamma}). \tag{38}$$

Similar to the derivation for GD, Eq. (35) controls the state transition, equivalent to Eq. (31). To simplify the further derivation of Eq. (36) and Eq. (37), we assume $\frac{\partial \boldsymbol{v}_{t+1}}{\partial \boldsymbol{g}_t} \approx \mathbf{0}$, which is reasonable

---

**Algorithm 2** PMP Solver for Adam

---

**Input:** LM learning rate $\eta$. Outer loop learning rate $\alpha$. Outer loop epochs $T_o$ Training data $\mathcal{D}$. Downstream loss function $J(\cdot)$. Training steps $T$. Function $\mathrm{Proj}[\cdot]$ that projects a point in $\mathbb{R}^D$ to the $D$-simplex.

**Output:** Data quality score $\boldsymbol{\gamma}^*$

$\boldsymbol{\gamma} = \left[\gamma_1, \gamma_2, \cdots, \gamma_{|\mathcal{D}|}\right] \leftarrow \left[\frac{1}{|\mathcal{D}|}, \frac{1}{|\mathcal{D}|}, \cdots, \frac{1}{|\mathcal{D}|}\right]; \boldsymbol{\Theta}_0 \leftarrow \left[\boldsymbol{\theta}_0^{(k)}, \mathbf{0}, \mathbf{0}\right]^\top$

   **repeat** $T_o$ **times**                                                                   ▷ Outer loop
      **for** $t = 0, 1, \cdots, T-1$ **do**                                         ▷ Forward inner loop
         $\boldsymbol{\Theta}_{t+1} \leftarrow \nabla_{\boldsymbol{\Lambda}} \mathcal{H}(\boldsymbol{\Theta}_t, \boldsymbol{\Lambda}_{t+1}, \boldsymbol{\gamma})$        ▷ Eq. (35), expanded to Eq. (32-33)
      **end for**
      $\boldsymbol{\Lambda}_T = [\nabla J(\boldsymbol{\theta}_T), \mathbf{0}, \mathbf{0}]^\top$
      **for** $t = T-1, T-2, \cdots, 1$ **do**                                    ▷ Reverse inner loop
         $\boldsymbol{\Lambda}_t \leftarrow \nabla_{\boldsymbol{\Theta}} \mathcal{H}(\boldsymbol{\Theta}_t, \boldsymbol{\Lambda}_{t+1}, \boldsymbol{\gamma})$        ▷ Eq. (36), expanded to Eq. (39-42)
      **end for**
      **for** $n = 1, 2, \cdots, |\mathcal{D}|$ **do**
         $\gamma_n \leftarrow \gamma_n + \alpha \nabla_{\gamma_n} \mathcal{H}(\boldsymbol{\Theta}_t, \boldsymbol{\Lambda}_{t+1}, \boldsymbol{\gamma})$         ▷ Eq. (37), expanded to Eq. (43)
      **end for**
      $\boldsymbol{\gamma} \leftarrow \mathrm{Proj}[\boldsymbol{\gamma}]$
   **end** and **return** $\boldsymbol{\gamma}$

---

because $\boldsymbol{v}_{t+1}$ is an exponential moving average of $\boldsymbol{g}_t^2$ and the weight $1 - \beta_2$ is usually much smaller than 1 in practice. Therefore, we have $\frac{\partial v_{t+1}}{\partial \boldsymbol{\theta}_t} \approx \mathbf{0}$, $\frac{\partial v_{t+1}}{\partial \boldsymbol{\gamma}_t} \approx \mathbf{0}$ and thus Eq. (36) can be written to

$$\boldsymbol{\Lambda}_t^* = \left[\boldsymbol{\Lambda}_t^{(1)}, \boldsymbol{\Lambda}_t^{(2)}, \boldsymbol{\Lambda}_t^{(3)}\right]^\top, \tag{39}$$

$$\boldsymbol{\Lambda}_t^{(1)} \approx \nabla J(\boldsymbol{\theta}_t^*) + \boldsymbol{\Lambda}_{t+1}^{(1)} - \frac{(1-\beta_1)\eta}{1-\beta_1^{t+1}} \nabla^2(\boldsymbol{\theta}_t^*, \boldsymbol{\gamma}^*) \frac{\boldsymbol{\Lambda}_{t+1}^{(1)}}{\sqrt{\widehat{\boldsymbol{v}}_{t+1}} + \epsilon}$$

$$+ \frac{(1-\beta_1)}{1-\beta_1^{t+1}} \nabla^2(\boldsymbol{\theta}_t^*, \boldsymbol{\gamma}^*) \boldsymbol{\Lambda}_{t+1}^{(2)}, \tag{40}$$

$$\boldsymbol{\Lambda}_t^{(2)} = -\frac{\eta\beta_1}{1-\beta_1^{t+1}} \frac{\boldsymbol{\Lambda}_{t+1}^{(1)}}{\sqrt{\widehat{\boldsymbol{v}}_{t+1}} + \epsilon} + \frac{\beta_1}{1-\beta_1^{t+1}} \boldsymbol{\Lambda}_{t+1}^{(2)}, \tag{41}$$

$$\boldsymbol{\Lambda}_t^{(3)} = \frac{\eta\beta_2}{1-\beta_2^{t+1}} \frac{\widehat{\boldsymbol{m}}_{t+1} \boldsymbol{\Lambda}_{t+1}^{(1)}}{\sqrt{\widehat{\boldsymbol{v}}_{t+1}} \left(\sqrt{\widehat{\boldsymbol{v}}_{t+1}} + \epsilon\right)^2} + \frac{\beta_2}{1-\beta_2^{t+1}} \boldsymbol{\Lambda}_{t+1}^{(3)}. \tag{42}$$

To achieve the minimum in Eq. (37), we need to compute the gradient of $\mathcal{H}$ with respect to $\boldsymbol{\gamma}_t$:

$$\nabla_{\gamma_n} \mathcal{H}(\boldsymbol{\Theta}_t^*, \boldsymbol{\Lambda}_{t+1}^*, \boldsymbol{\gamma}) = \frac{\eta(1-\beta_1)}{1-\beta_1^{t+1}} \boldsymbol{\Lambda}_{t+1}^{(1)}{}^\top \frac{\nabla l(x_n, \boldsymbol{\theta}_t)}{\sqrt{\widehat{\boldsymbol{v}}_{t+1}} + \epsilon} + \frac{1-\beta_1}{1-\beta_1^{t+1}} \boldsymbol{\Lambda}_{t+1}^{(2)}{}^\top \nabla l(x_n, \boldsymbol{\theta}_t) \tag{43}$$

In this way, we can use Algorithm 2 to solve for the data quality scores on the proxy dataset $\mathcal{D}^{\mathrm{prx}}$, just like Algorithm 1 in Section 2.3.1. Then, we train a data scorer with the solved data weights $\boldsymbol{\gamma}^*$ as in Section 2.3.2 and conduct data selection based on the scores inferred by the data scorer on $\mathcal{D}$ as in Section 2.3.3. In pilot experiments, we find that computing data quality scores based on Adam does not show substantial improvement over that based on GD. Given that Algorithm 2 requires twice as much computation and 3 times as much GPU memory as Algorithm 1, we adopt PMP condition for data selection based on GD (Theorem 2.1) to build PDS in our main paper.

## D    ALGORITHM 1 AS PROXIMAL GRADIENT DECENT

We provide another view of Algorithm 1 as using Proximal Gradient Decent method (Bauschke & Combettes, 2011). Specifically, we can optimize Eq. (3) with the following rules:

$$A(\boldsymbol{\gamma}) = \sum_{t'=1}^{T} J(\boldsymbol{\theta}_{t'}), \tag{44}$$

$$\boldsymbol{\gamma} \leftarrow \mathrm{Proj}\left[\boldsymbol{\gamma} - \alpha' \nabla_{\boldsymbol{\gamma}} A\right], \tag{45}$$

where $A(\boldsymbol{\gamma})$ denotes the cost function in Eq. (3), $\alpha'$ is the learning rate and $\mathrm{Proj}[\cdot]$ projects a point in $\mathbb{R}^D$ to the $D$-simplex. $\nabla_{\boldsymbol{\gamma}} A = \left[\frac{\partial A}{\partial \gamma_1}, \frac{\partial A}{\mathrm{d}\gamma_2}, \cdots, \frac{\partial A}{\partial \gamma_D}\right]$ is the gradient of $A(\boldsymbol{\gamma})$ with respect to the data weights $\boldsymbol{\gamma}$. Now, we compute each element of $\nabla_{\boldsymbol{\gamma}} A$ with the chain rule:

$$
\begin{aligned}
\frac{\partial A}{\partial \gamma_n} &= \sum_{t'=1}^{T} \frac{\partial J(\boldsymbol{\theta}_{t'})}{\partial \gamma_n} \\
&= \sum_{t'=1}^{T} \nabla J(\boldsymbol{\theta}_{t'})^{\top} \frac{\partial \boldsymbol{\theta}_{t'}}{\partial \gamma_n} \\
&= \sum_{t'=1}^{T} \nabla J(\boldsymbol{\theta}_{t'})^{\top} \sum_{t=1}^{t'} \frac{\partial \boldsymbol{\theta}_{t'}}{\partial \boldsymbol{\theta}_t} \frac{\partial \boldsymbol{\theta}_t}{\partial \gamma_n} \\
&= -\eta \sum_{t'=1}^{T} \nabla J(\boldsymbol{\theta}_{t'})^{\top} \sum_{t=1}^{t'} \frac{\partial \boldsymbol{\theta}_{t'}}{\partial \boldsymbol{\theta}_t} \nabla l(x_n, \boldsymbol{\theta}_{t-1}) \\
&= -\eta \sum_{t'=1}^{T} \sum_{t=0}^{t'-1} \nabla J(\boldsymbol{\theta}_{t'})^{\top} \frac{\partial \boldsymbol{\theta}_{t'}}{\partial \boldsymbol{\theta}_{t+1}} \nabla l(x_n, \boldsymbol{\theta}_t) \\
&= -\eta \sum_{t=0}^{T-1} \left[\sum_{t'=t+1}^{T} \nabla J(\boldsymbol{\theta}_{t'})^{\top} \frac{\partial \boldsymbol{\theta}_{t'}}{\partial \boldsymbol{\theta}_{t+1}}\right] \nabla l(x_n, \boldsymbol{\theta}_t),
\end{aligned}
\tag{46}
$$

where $\frac{\partial \boldsymbol{\theta}_{t'}}{\partial \boldsymbol{\theta}_{t+1}}$ satisfies

$$\frac{\partial \boldsymbol{\theta}_{t'}}{\partial \boldsymbol{\theta}_t} = \frac{\partial \boldsymbol{\theta}_{t+1}}{\partial \boldsymbol{\theta}_t} \frac{\partial \boldsymbol{\theta}_{t'}}{\partial \boldsymbol{\theta}_{t+1}} = \left[\boldsymbol{I} - \eta \nabla^2 L(\boldsymbol{\theta}_t, \boldsymbol{\gamma}_t)\right] \frac{\partial \boldsymbol{\theta}_{t'}}{\partial \boldsymbol{\theta}_{t+1}}, \tag{47}$$

according to Eq. (2). In the following, we show that:

$$\boldsymbol{\lambda}_{t+1} = \sum_{t'=t+1}^{T} \frac{\partial \boldsymbol{\theta}_{t'}}{\partial \boldsymbol{\theta}_{t+1}} \nabla J(\boldsymbol{\theta}_{t'}), \tag{48}$$

where $\boldsymbol{\lambda}_{t+1}$ is the co-state vector in Algorithm 1. Let $\boldsymbol{\lambda}'_{t+1} = \sum_{t'=t+1}^{T} \frac{\partial \boldsymbol{\theta}_{t'}}{\partial \boldsymbol{\theta}_{t+1}} \nabla J(\boldsymbol{\theta}_{t'})$, we then have $\boldsymbol{\lambda}'_T = \nabla J(\boldsymbol{\theta}_T)$ and the following difference equation for $\boldsymbol{\lambda}'_t$:

$$
\begin{aligned}
\boldsymbol{\lambda}'_t &= \sum_{t'=t}^{T} \frac{\partial \boldsymbol{\theta}_{t'}}{\partial \boldsymbol{\theta}_t} \nabla J(\boldsymbol{\theta}_{t'}) \\
&= \nabla J(\boldsymbol{\theta}_{t'}) + \sum_{t'=t+1}^{T} \frac{\partial \boldsymbol{\theta}_{t'}}{\partial \boldsymbol{\theta}_t} \nabla J(\boldsymbol{\theta}_{t'}) \\
&= \nabla J(\boldsymbol{\theta}_{t'}) + \sum_{t'=t+1}^{T} \left[\boldsymbol{I} - \eta \nabla^2 L(\boldsymbol{\theta}_t, \boldsymbol{\gamma})\right] \frac{\partial \boldsymbol{\theta}_{t'}}{\partial \boldsymbol{\theta}_{t+1}} \nabla J(\boldsymbol{\theta}_{t'}) \\
&= \nabla J(\boldsymbol{\theta}_{t'}) + \boldsymbol{\lambda}'_{t+1} - \eta \nabla^2 L(\boldsymbol{\theta}_t, \boldsymbol{\gamma}) \boldsymbol{\lambda}'_{t+1}.
\end{aligned}
\tag{49}
$$

Given that $\boldsymbol{\lambda}'_{t+1}$ satisfies the same difference equation as $\boldsymbol{\lambda}_{t+1}$ in Algorithm 1 and has the same value at $t = T$, we have $\boldsymbol{\lambda}'_{t+1} = \boldsymbol{\lambda}_{t+1}$. Therefore, the gradient in Eq. (46) can be written as

$$\frac{\partial J}{\partial \gamma_n} = -\eta \boldsymbol{\lambda}_{t+1}^\top \nabla l(x_n, \boldsymbol{\theta}_t). \tag{50}$$

Combining Eq. (45) and Eq. (50), we recover the update rules of $\gamma_{n,t}$ in Algorithm 1 by setting $\alpha' = \alpha / \eta$.

## E  EXTENSION TO CONSIDERING DATA DEPENDENCE

In Section 2.1, we formulate our problem by focusing on the individual importance of each data point to the downstream loss. This potentially affects the performance of PDS when the original corpus to select from contain too much duplication. The reason is that, similar examples tend to have similar distances to $\boldsymbol{\lambda}$, which means that if one example individually get high $\gamma$, the others will also get high $\gamma$ and be selected, affecting the diversity of the selected corpus. In this section, we provide a discussion on how the dependence between the selection data points can be considered to improve the diversity of the selected data, as an extension to our theoretical framework. Specifically, we can improve the pair-wise difference between the examples to promote the diversity of the selected data by considering the following pair-wise similarity:

$$\text{sim}(x_n, x_m) = \frac{\boldsymbol{h}_n^\top \boldsymbol{h}_m}{||\boldsymbol{h}_n||||\boldsymbol{h}_m||}, \tag{51}$$

where $\boldsymbol{h}_n$ and $\boldsymbol{h}_m$ are the representations of each example generated by a model like RoBERTa (Liu et al., 2019). Then, we have the following loss to minimize, which encourages the difference between examples:

$$\mathcal{L}^{\text{diversity}} = \sum_{n,m} \text{sim}(x_n, x_m) \gamma_n \gamma_m = \boldsymbol{\gamma}^\top \boldsymbol{S} \boldsymbol{\gamma}, \tag{52}$$

where $\boldsymbol{S} = [\text{sim}(x_n, x_m)]_{1 \leq n \leq |\mathcal{D}|, 1 \leq m \leq |\mathcal{D}|}$. We can add this loss to the optimization problem in Eq. (3), using a hyper-parameter $w$ to control its weight. Additionally, is it also useful to add an $l_0$-norm constraint on $\gamma$ to constrain the number of non-zero elements to the selected data number $K$. This is because when pairwise interactions between examples are considered, adding or removing one data point would affect the contribution of the others. Therefore, the optimization problem becomes:

$$\min_{\boldsymbol{\gamma}} \sum_{t=1}^{T} J(\boldsymbol{\theta}_t) + w \cdot \boldsymbol{\gamma} \boldsymbol{S} \boldsymbol{\gamma}$$
$$\text{s.t. } \boldsymbol{\theta}_{t+1} = \boldsymbol{\theta}_t - \eta \nabla L(\boldsymbol{\theta}_t, \boldsymbol{\gamma}), \;\; \boldsymbol{\gamma} \in U, \;\; \sum_{n=1}^{N} \mathbb{1}[\gamma_n > 0] = K. \tag{53}$$

However, the problem with the $l_0$-norm constraint is computationally intractable to solve because computing the $l_0$-norm involves determining the exact sparsity pattern, which is equivalent to solving a combinatorial optimization problem. For high-dimensional $\boldsymbol{\gamma}$, this becomes computationally prohibitive as the complexity grows exponentially with the size of the problem. Furthermore, it is unclear how the constraint would affect the physical property of the dynamics and resulting in the optimization problem. For example, whether it will introduce any singularity during the dynamic process or the inaccuracy from the approximated optimization procedure (given $l_0$-norm is not differentiable) would lead to any unexpected outcome. We leave the design of effective and efficient methods to solve the problem in Eq. (53) to future work.

## F  DISCUSSION ON EQ. (6) OF THEOREM 2.1

**The Optimality.** In Theorem 2.1, Eq. (6) takes a maximum operation to get $\boldsymbol{\gamma}^*$, which increases the $\gamma$ scores with the highest $\sum_{t=0}^{T} \lambda_{t+1}^\top \nabla l(x_n, \theta^*)$ values and reduce those $\gamma$ with lower $\sum_{t=0}^{T} \lambda_{t+1}^\top \nabla l(x_n, \theta^*)$ values to 0. However, **this does not imply that a single training example is the optimal for data selection** for the following reasons:

- First, there could exist several samples having the same largest $\sum_{t=0}^{T} \lambda_{t+1}^{\top} \nabla l(x_n, \theta^*)$ values when the model's parameter $\theta^*$ is trained under the optimal $\gamma$ scores, i.e. $\exists S = \{n_1, n_2, \cdots, n_{|S|}\}$ such that for any $m \notin S$.

$$\sum_{t=0}^{T} \lambda_{t+1}^{\top} \nabla l(x_{n_1}, \theta^*) = \cdots = \sum_{t=0}^{T} \lambda_{t+1}^{\top} \nabla l(x_{n_{|S|}}, \theta^*) > \sum_{t=0}^{T} \lambda_{t+1}^{\top} \nabla l(x_m, \theta^*) \qquad (54)$$

Therefore, multiple samples in $S$ can have non-zero $\gamma^*$ scores to satisfy Eq. (6), as long as those samples not in $S$ get zero $\gamma^*$ scores. Accordingly, although "a single training sample" is a solution to Eq. (6), **there also exist many more "equally qualified" solutions (such as assigning uniform $\gamma$ values to the data points in $S$) that satisfy Eq. (6).**

- Second, Theorem 2.1 is a necessary condition of the $\gamma$'s optimality. This is because the two techniques we used to prove Theorem 2.1 in Appendix B, i.e., (1) time-variant PMP for discrete dynamical system (Theorem B.1) and (2) The Lagrange Multiplier method, are all necessary conditions. Therefore, the unreasonable "a single training sample" is not necessarily the optimal solution to the problem.

Many empirical results (Geering, 2007; Seierstad & Sydsaeter, 1986) have shown that solving this necessary condition can successfully lead to fairly good solutions, which is much like using gradient descent in deep learning even if it does not guarantee the optimality of the solution.

**Intuition Behind $\gamma$ Updates.** During the update of $\gamma$ in Algorithm 1, since $\theta$ has not reached its optimum yet, the sample set $S$ having the same largest $\sum_{t=0}^{T} \lambda_{t+1}^{\top} \nabla l(x_n, \theta)$ values has not emerged, making the max operation unstable in practice. Therefore, we increase $\gamma$ by a value proportional to $\sum_{t=0}^{T} \lambda_{t+1}^{\top} \nabla l(x_n, \theta)$ in each iteration of the outer loop as in Algorithm 1. This updating strategy still satisfies our theory because it guarantees that the sample set having the largest $\sum_{t=0}^{T} \lambda_{t+1}^{\top} \nabla l(x_n, \theta)$ values gets the highest $\gamma$ values. **It also ensures that the samples with the same $\sum_{t=0}^{T} \lambda_{t+1}^{\top} \nabla l(x_n, \theta)$ values receive the same $\gamma$ scores, which avoids the "single training sample" solution.**

## G  IMPLEMENTATION DETAILS

### G.1  SOLVING DATA QUALITY SCORES

Algorithm 1 needs to compute the Hessian matrix, as in Eq. (5), and per-instance vector-gradient product, as in Eq. (6). We adopt the Jacobian-Vector-Product[4] in PyTorch (Paszke et al., 2019) to efficiently implement these operations. There is still a large room for this implementation, such as adopting other deep learning frameworks (Dagréou et al., 2024; Bradbury et al., 2018) or using lower-complexity algorithms (Wang et al., 2024). Algorithm 1 requires storing all the LM checkpoints $\theta_t$ from $t = 0$ to $t = T - 1$ in the forward inner loop for computing $\lambda_t$ in the reverse inner loop. To reduce the single-device GPU memory use, we implement an activation partition algorithm inspired by ZeRO-2 (Rajbhandari et al., 2020), where $\theta_t$ in one inner-loop pass are stored in different GPU devices. We also adopt a strategy inspired by activation checkpointing (Chen et al., 2016).

### G.2  TRAINING DATA SCORER

We fine-tune the Fairseq-Dense-125M model (Artetxe et al., 2022) on the solved data weights $\gamma^*$ to obtain the data scorer. as in Eq. (8), we adopt a linear transformation on the mean pooling of the instance's representations along the sequence length. The size of the hidden state is 768. We optimize Eq. (8) with the AdamW (Loshchilov & Hutter, 2019) optimizer for 5 epochs, using a learning rate $1 \times 10^{-4}$ and a batch size 512. We split 10% samples from $\mathcal{D}^{\mathrm{prx}}$ for validation and select the checkpoint achieving the highest Spearman correlation score on the validation set to infer data quality scores in $\mathcal{D}$.

---

[4] https://pytorch.org/docs/stable/func.api.html

| Model Size | $d_{\text{model}}$ | $d_{\text{FFN}}$ | $n_{\text{layers}}$ | $n_{\text{head}}$ | $d_{\text{head}}$ | learning rate |
|---|---|---|---|---|---|---|
| 160M | 768 | 3,072 | 12 | 12 | 64 | $6 \times 10^{-4}$ |
| 470M | 1,024 | 4,096 | 24 | 16 | 64 | $3 \times 10^{-4}$ |
| 1B | 1,536 | 6,144 | 24 | 16 | 96 | $2.5 \times 10^{-4}$ |
| 1.7B | 2,048 | 8,192 | 24 | 16 | 128 | $2 \times 10^{-4}$ |

Table 6: Model configurations and corresponding learning rates.

### G.3 PRE-TRAINING

We pre-train all our models for about 50B tokens with a batch size of 512 and a max input length of 1,024. We use the AdamW (Loshchilov & Hutter, 2019) optimizer and cosine learning rate scheduler, which warmups the learning rates for 2K steps and then decays it to 10% of the maximal value. We follow (Brown et al., 2020) to set the maximal learning rates as listed in Table 6, together with the model configurations.

### G.4 EVALUATION

We evaluate our trained models on MMLU (Hendrycks et al., 2021) and the evaluation sets used in OLMo (Groeneveld et al., 2024), including Hellaswag (HS; Zellers et al., 2019), Winograde (Wino.; Levesque et al., 2012), LAMBADA (LAMB; Paperno et al., 2016), OpenbookQA (OBQA; Mihaylov et al., 2018), ARC-easy/challenge (ARC-e/c; Clark et al., 2018), PIQA (Bisk et al., 2020), SciQ (Welbl et al., 2017), and BoolQ (Clark et al., 2019). We adopt the LM-evaluation-harness library (Gao et al., 2024)[5] to conduct the zero-shot evaluation, which formulates the tasks as the multiple-choice problems and chooses the answer by comparing the answer-length-normed loss across candidates (`acc_norm`). We also compute the test loss of our trained models on the DCLM corpus (Li et al., 2024), a 10K subset uniformly sampled from the high-quality pre-training corpus in Li et al. (2024).

### G.5 BASELINES

**Reducible Holdout Loss Selection (RHO-Loss; Mindermann et al., 2022; Lin et al., 2024).** RHO-Loss selects data with high "reducible holdout loss" defined as follows:

$$\text{RHO-Loss}_n = l(x_n, \boldsymbol{\theta}_T) - l(x_n, \boldsymbol{\theta}^{\text{down}}), \tag{55}$$

where $\boldsymbol{\theta}^{\text{down}}$ is the model parameters trained on the downstream tasks, which is LIMA (Zhou et al., 2024) in our experiments. We first split 10% of LIMA for validation and choose $\boldsymbol{\theta}^{\text{down}}$ with the lowest validation loss. Then, we train $\boldsymbol{\theta}_t$ using $x_n$ with the top $\alpha \times 100\%$ RHO-Loss$_n$. We tried $\alpha \in \{0.1, 0.3, 0.5, 0.7\}$ and find that $\alpha = 0.5$ performs the best.

**Data Selection via Importance Resampling (DSIR; Xie et al., 2023).** DSIR selects pre-training data based on the n-gram feature overlap between the instances in the downstream dataset (LIMA (Zhou et al., 2024) in our experiments) and the large-scale corpus. Pre-training instances whose features have high probabilities under the feature distribution of the downstream dataset will obtain higher sampling weights. We use the official implementation of DSIR[6].

**Influence Score (IF-Score Koh & Liang, 2017; Grosse et al., 2023).** We adopt the influence function (Koh & Liang, 2017; Grosse et al., 2023) to measure the quality of $x$ using the downstream loss $J(\boldsymbol{\theta})$ as the target:

$$\text{IF-Score}(x) = \nabla l(x, \boldsymbol{\theta}_T)^{\top} \left[ \nabla^2 L(\boldsymbol{\theta}_T) \right]^{-1} \nabla J(\boldsymbol{\theta}_T), \tag{56}$$

where $\boldsymbol{\theta}_T$ is the LM parameters trained for $T$ steps and $L(\boldsymbol{\theta}_T) = \frac{1}{|\mathcal{D}|} \sum_{x \in \mathcal{D}} l(x, \boldsymbol{\theta}_T)$. We adopt a linear-time iterative algorithm to compute the inverse-Hessian-vector-product (Agarwal et al., 2017).

---

[5] https://github.com/t1101675/lm-harness/
[6] https://github.com/p-lambda/dsir

| | HS | LAMB | Wino. | OBQA | ARC-e | ARC-c | PIQA | SciQ | BoolQ | Avg. |
|---|---|---|---|---|---|---|---|---|---|---|
| Conventional | 32.2 | 34.9 | 51.4 | 25.6 | 40.9 | 22.5 | 58.3 | 65.5 | 57.6 | 43.2 |
| RHO-Loss | 32.2 | 35.3 | **53.2** | 28.1 | 40.5 | **24.1** | 58.5 | 63.1 | 53.0 | 43.1 |
| DSIR | 32.8 | 35.7 | 52.5 | 26.6 | 41.2 | 23.8 | 57.8 | 68.7 | 54.7 | 43.8 |
| IF-Score | 32.2 | 35.7 | 51.1 | 27.4 | 40.8 | 22.6 | 57.6 | 64.1 | 55.8 | 43.0 |
| PDS | **33.5** | **38.2** | 51.4 | **28.4** | **42.3** | **24.1** | **59.2** | **68.8** | **58.7** | **45.0** |

Table 7: Downstream evaluation results on 160M models. We report the accuracy scores and the average scores across the datasets. The best scores of each model size are **boldfaced**. PDS achieves the best performance in most cases compared to the baselines.

To reduce the computational cost, we adopt a similar efficient implementation as PDS by computing the IF-Scores on a small proxy data based on a small proxy LM and then transferring these scores to a large pre-training corpus with a fine-tuned data scorer. We select examples with the top 40% scores inferred by the data scorer on the large pre-training corpus.

## G.6 SIMULATED SETTING FOR EXPERIMENTS IN TABLE 6

To exactly run Algorithm 1 with a feasible computational overhead, we adopt a 12M LM from the Mistral (Jiang et al., 2023) family, with a small vocabulary. We uniformly sample 4,096 instances as $\mathcal{D}$, with a max sequence length of 256, and construct a 16K vocabulary from $\mathcal{D}$ and LIMA. We run the inner loops for 5K steps and the outer loop for 5 epochs to get the PDS (exact) line in Figure 6. For PDS (Efficient), we adopt an 8.7M model as the proxy LM and set $M = 5$, $T^{\mathrm{prx}} = 100$. We run the inner loop using SGD with a batch size of 128. The outer loop epoch number is set to 1.

## H COMPLEXITY ANALYSIS

Following Hoffmann et al. (2022), for an LM with $N$ parameters to be trained on $D$ tokens, we assume the computational FLOPs of a forward and a backward pass are $2ND$ and $4ND$, respectively. We compute the FLOPs and asymptotic complexities of different stages in PDS as follows:

- **Solving Data Quality Scores**: According to Section 2.3.1, we first pre-train a proxy LM on $\mathcal{D}$ which consumes $6N^{\mathrm{prx}}D$ FLOPs. Then, we perform Algorithm 1 $M$ times on $\mathcal{D}^{\mathrm{prx}}$ based on the proxy LM. The forward inner loop in Algorithm 1 consumes $6N^{\mathrm{prx}}D^{\mathrm{prx}}$ FLOPs. The reverse inner loop can be treated as the "backward" propagation of the forward inner loop as discussed in Appendix D, which consumes $2 \times 6N^{\mathrm{prx}}D^{\mathrm{prx}}$ FLOPs. The update of $\gamma$ results in one forward and backward pass of the proxy LM on $\mathcal{D}^{\mathrm{prx}}$, which consumes $6N^{\mathrm{prx}}D^{\mathrm{prx}}$ FLOPs. In summary, the asymptotic complexity of solving data quality scores is $O(N^{\mathrm{prx}}D + 4MN^{\mathrm{prx}}D^{\mathrm{prx}})$.

- **Data Scorer**: The data scorer with $N^{\mathrm{score}}$ is trained on $\mathcal{D}^{\mathrm{prx}}$ and used to infer data quality scores on $\mathcal{D}$. Therefore, the computation overhead is $6N^{\mathrm{score}}\mathcal{D}^{\mathrm{prx}} + 2N^{\mathrm{score}}\mathcal{D}$ and the asymptotic complexity is $O(3N^{\mathrm{score}}\mathcal{D}^{\mathrm{prx}} + N^{\mathrm{score}}\mathcal{D})$.

- **Data Selection**: Selecting pre-training corpus requires iterating over $\mathcal{D}$, whose asymptotic complexity is $O(D)$. This process can be done on CPUs and does not require GPU FLOPs.

- **Pre-Training**: Pre-training an LM with $N$ parameters requires $6ND$ FLOPs, whose asymptotic complexity is $O(ND)$.

## I MORE RESULTS

### I.1 RESULTS ON 160M MODELS

We present the results on the OLMo (Groeneveld et al., 2024) evaluation sets based on the 160M models in Table 7. PDS achieves the best performance in most cases compared to the baselines.

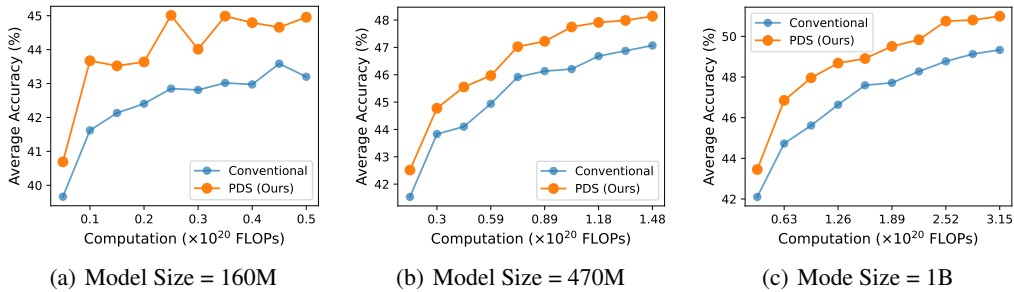

(a) Model Size = 160M      (b) Model Size = 470M      (c) Mode Size = 1B

Figure 8: Scaling curves of average accuracy on the OLMo (Groeneveld et al., 2024) evaluation datasets with respect to computation for 160M, 470M, and 1B sizes.

| Method | PPL (160M) | PPL (470M) |
|---|---|---|
| Conventional | 44.7 | 34.8 |
| RHO-Loss | 42.1 | 33.0 |
| DSIR | 40.5 | 34.0 |
| IF-Score | 41.5 | 31.1 |
| PDS | **36.6** | **27.1** |

Table 8: The LMs' perplexity (PPL) on the ground truth for zero-shot evaluation on MMLU (Hendrycks et al., 2021). The best scores of each model size are **boldfaced**.

### I.2 SCALING CURVES OF COMPUTATION FOR OTHER MODEL SIZES

We plot scaling curves of computation for 160M, 470M, and 1B models in Figure 8. PDS-selected data accelerates the model learning across model sizes.

### I.3 RESULTS ON MMLU MEASURED BY PERPLEXITY ON GROUND TRUTH

As suggest by Schaeffer et al. (2023), using more smooth and continuous metrics like the LM's perplexity on the ground truth better reveals the gap between different base models. Therefore, we include this results in Table 8 as a supplement to Table 2, which shows substantial improvement of PDS over the baselines.

### I.4 TEST LOSS EXTRAPOLATION WITH THE SCALING LAW

We extrapolate the test losses on the DCLM corpus (Li et al., 2024) of the conventionally trained and PDS-trained LMs with the Scaling Law (Hoffmann et al., 2022; Kaplan et al., 2020). Following Hoffmann et al. (2022), we consider the scaling law with the following form:

$$L(N, D) = E + \frac{A}{N^\alpha} + \frac{B}{D^\beta}, \tag{57}$$

where $N$ is the model parameters, $D$ is the trained tokens, and $A$, $B$, $E$, $\alpha$, $\beta$ are constants. We obtain these constants by minimizing the Huber loss (Huber, 1992):

$$\min_{a,b,e,\alpha,\beta} \sum_{(N_i, D_i, L_i)} \text{Huber}_\delta(\text{LSE}(a - \alpha \log N_i, b - \beta \log D_i, e) - \log L_i), \tag{58}$$

where $\text{LSE}(\cdot)$ is the log-sum-exp operation. The loss is summed over all $(N_i, D_i, L_i)$, which is obtained by the test losses of 160M, 470M, 1B, and 1.7B LM during training from 0B to 50B tokens. We record the losses every 2.5B tokens, resulting in a total $4 \times 50B/2.5B = 80$ tuples like $(N_i, D_i, L_i)$. After solving $a$, $b$, and $e$ from Eq. (58), we have $A = \exp(a)$, $B = \exp(b)$, and $E = \exp(e)$.

|  | $A$ | $B$ | $\alpha$ | $\beta$ | $E$ |
|---|---|---|---|---|---|
| Conventional | $8.09 \times 10^2$ | $7.50 \times 10^5$ | 0.397 | 0.651 | 2.829 |
| PDS | $6.21 \times 10^3$ | $1.76 \times 10^5$ | 0.518 | 0.585 | 2.829 |

Figure 9: Scaling law constants by fitting the test losses on the DCLM corpus.

|  | $N =$160M | $N =$470M | $N =$1B | $N =$1.7B | $D = 50 \times 10^9$ |
|---|---|---|---|---|---|
| Conventional | 0.990 | 0.998 | 0.993 | 0.996 | 0.999 |
| PDS | 0.992 | 0.995 | 0.997 | 0.993 | 0.998 |

Table 9: Correlations ($R^2$) of fitting the scaling curves.

Hoffmann et al. (2022) optimizes Eq. (58) using the LBFGS algorithm (Liu & Nocedal, 1989). However, we find this algorithm sensitive to the initialization of the parameters to be optimized. Therefore, we apply a two-stage optimization. Specifically, we first fit the following data scaling curves for $N =$160M, 470M, 1B, and $1.7B$ with non-linear least squares from `scipy.optimize.curve_fit`[7], which is much more robust to the initialization:

$$L(D) = E'(N) + \frac{B_0(N)}{D^{\beta_0(N)}}, \tag{59}$$

where $E'(N)$, $B_0(N)$ and $\beta_0(N)$ are the fitted parameters. Then, we fit the following model size scaling curve:

$$E' = E_0 + \frac{A_0}{N^{\alpha_0}}. \tag{60}$$

We use the constants from Eq. (60) and the average constants from Eq. (59) to compute the initialization for the LBFGS algorithm:

$$
\begin{aligned}
a_0 &= \log A_0, \\
b_0 &= \log \frac{B_0(160M) + B_0(470M) + B_0(1B) + B_0(1.7B)}{4}, \\
\alpha_0 &= \alpha_0, \\
\beta_0 &= \frac{\beta_0(160M) + \beta_0(470M) + \beta_0(1B) + \beta_0(1.7B)}{4}, \\
e_0 &= \log E_0,
\end{aligned}
\tag{61}
$$

where $a_0, b_0, \alpha_0, \beta_0, e_0$ are the parameter initialization for the LFBGS algorithm to optimize Eq. (58). We set $\delta = 1 \times 10^{-3}$ and learning rate to 0.05 when running LFBGS and obtain the constants in Table 9. We use these constants and Eq. (57) to compute the predicted loss in Table 3.

In Section 3.3 (Data-Constrained Setting), to compute the expected token demand of conventional pre-training to match the performance of PDS ($r = 0.25$, 4 Eps.), we solve for $D$ using the constants in Table 3 and use $\frac{D}{4}$ as the token demand, indicating the LM can be trained for 4 epochs as suggested by Muennighoff et al. (2023).

**Goodness of Fit.** We evaluate the goodness of fit of the scaling curves with respect to the training token size $D$ and model size $N$ respectively, by computing the correlation coefficient $R^2 = 1 - \frac{\sum_i (y_i - \hat{y}_i)^2}{\sum_i (y_i - \overline{y})^2}$, where $y_i$ is the ground truth value and $\hat{y}_i$ is the prediction.

Regarding the training token size, to make the original problem a linear regression problem, we convert Eq. (57) into

$$\log \left( L(N, D) - E - \frac{A}{N^\alpha} \right) = \log B - \beta \log D. \tag{62}$$

---

[7]https://docs.scipy.org/doc/scipy/reference/generated/scipy.optimize.curve_fit.html

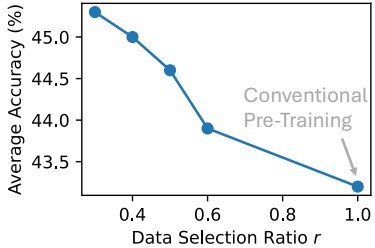 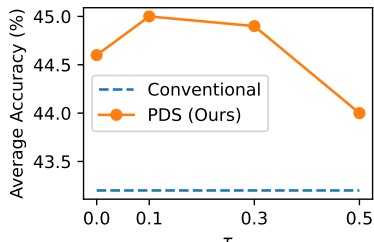

Figure 10: Effect of the data selection ratio $\alpha$. We report the average accuracy on the OLMo evaluation datasets for $\alpha \in [0.3, 0.4, 0.5, 0.6, 1.0]$.

Figure 11: Effect of the strength $\tau$ in Gumble-Top-$K$. We report the average accuracy on the OLMo evaluation datasets for $\tau \in [0.0, 0.1, 0.3, 0.5]$.

Then, we consider $\log \left( l_i - E - \frac{A}{N^\alpha} \right)$ as the ground truth value for regression, where $l_i$ is the observed loss, and $\log B - \beta \log D_i$ as the prediction. For each $N \in [160M, 470M, 1B, 1.7B]$ we compute an $R^2$ respectively. Similarly, regarding the model size, we convert Eq. (57) into

$$\log \left( L(N, D) - E - \frac{B}{D^\beta} \right) = \log A - \alpha \log N, \tag{63}$$

to compute the corresponding $R^2$ that measures its goodness of fit. For simplicity, we only consider the models at the end of training, where $D = 50 \times 10^9$. The results in Table I.4 show that the correlations are sufficiently high, suggesting the scaling curve fits the impact from both the data and model sizes very well.

## I.5 ABLATION STUDIES

**Data Selection Ratio.** In Figure 10, we investigate how the data selection ratio affects the performance of PDS when the original training corpus is sufficiently large (in Section 3.3, we explore the data selection ratio in the data-constrained setting.). A lower data selection ratio results in better final model performance. However, to ensure that the selected data contains enough training tokens, a larger original corpus is needed for lower $\alpha$. Therefore, we keep $\alpha = 0.4$ in our main experiments to balance effectiveness and data demand.

**Gumbel Noise in Data Selection** In Figure 11, we explore the effect of the strength $\tau$ in Gumble-Top-$K$ used for data selection in Eq. (9). We can see that $\tau = 0.1$ achieves the best performance, verifying the benefits of increasing the diversity of the pre-training corpus. Too large a $\tau$ value makes PDS degenerate to random selection, causing the performance to decrease.

**Choice of $J(\boldsymbol{\theta})$.** The desired data plays an important role in determining the quality of the selected data. We test different downstream datasets to compute $J(\boldsymbol{\theta})$ in PDS and report the model performance in Table 10. The comparison between the results of using LAMBADA and LIMA shows the importance of the downstream data diversity. Instances in LAMBADA mostly come from stories, while LIMA is composed of instruction-response pairs that cover various tasks, which yields better overall performance. When comparing LIMA, OpenWebText Gokaslan et al. (2019), and CC, we conclude that data quality is another major concern. Although Open-WebText has been shown to have better scaling factors (Bi et al., 2024) and used as the target set (Brown et al., 2020), replacing it with higher quality LIMA further improves performance. Compared to diversity and quality, large sizes of downstream datasets seem less important, because LIMA performs the best with the least instance number.

|  | $J(\boldsymbol{\theta})$ | Acc. |
|---|---|---|
| Conventional | - | 43.2 |
| PDS | LAMB. | 43.7 |
|  | CC | 43.0 |
|  | OWT | 44.1 |
|  | LIMA | 45.0 |

Table 10: Effect of using different downstream datasets to compute $J(\boldsymbol{\theta})$. We report average accuracy on the OLMo evaluation datasets.

