# OpenReview forum: "Data Selection via Optimal Control for Language Models"
_ICLR.cc/2025/Conference — ICLR 2025 Oral_

### Official Review · Reviewer_NWbj · 2024-10-22

**Soundness:** 3
**Presentation:** 4
**Contribution:** 3
**Rating:** 10
**Confidence:** 2

**Summary:**

This paper presents a formulation of turning the data selection to the AUC optimization problem. More importantly, this paper derives the optimal solution for the data selection problem and proposes an efficient implementation of the proposed method. The proposed methods are empirically verified in diverse settings.

**Strengths:**

This work is sufficiently solid. It covers the fundamental problem formulation, which introduces a new optimization problem. Also, it provides the theoretical optimization solution to the proposed problem, which has not been studied before. From this perspective, this paper has made sufficiently novel contributions.  Moreover, this work designs an efficient implementation to avoid additional computation overhead. These methods are empirically verified with different experiments setting, demonstrating the effectiveness of this method across different settings. Due to its theoretical contributions and solid experiments, I give a clear accept (score 10) to this paper.

**Weaknesses:**

I didn't identify any major weaknesses of this paper.

**Questions:**

It is really interesting to see the formulation of optimization problem (3) in the data selection problem. Has this method been used in other tasks such as computer vision, diffusion model, or simple MNIST dataset classification? It seems to be a very general approach that can be used in many senarios.

In Line 60, the PMP derives the necessary condition for optimal data selection. Does it mean that it is not sufficient to guarantee the data selection is optimal?

It seems that the PDS method always achieves better performance. Is there any negative results where the PDS method fails? Do we have any senarios where we should avoid using this method for the data selection?

---

> ### Author Response · Authors · 2024-11-21
> **Response to Reviewer NWbj**
>
> We are deeply grateful for the reviewer's highly positive feedback and  recognizing the novelty and significance of our contributions. The reviewer's acknowledgment of our theoretical formulation, optimization solution, efficient implementation, and robust experimental validation is really encouraging. We are especially honored by the reviewer's clear and strong support!
>
> ## The use of our formulation in other scenarios (question #1)
>
> To our best knowledge, there is no existing work applying this formulation in data selection. We believe that it is quite promising to apply this formulation to  multi-modal foundation model training, such as vision language models, because we do not make any assumptions of the data forms or its content. In our on-goning project, we apply PDS to select image-text pairs for SFT of vision language models and observed promising results.
>
> ## The necessary condition (question #2)
>
> Yes, the PMP conditions do not guarantee that the data selection is optimal, since they are necessary conditions. However, according to many empirical results in the field of Optimal Control [1,2], solving the PMP conditions is sufficient to get fairly good solutions. (Much like using Gradient Descent in deep learning even if Gradient Decent also cannot guarantee the global optimality.)
>
> [1] Optimal Control with Engineering Applications. 2007. Springer.
>
> [2] Optimal Control Theory with Economic Applications. 1987. Amsterdam: North-Holland.
>
> ## The potential negative results (question #3)
>
> A potential scenario that PDS may fail is that the original corpus contains too much duplication. Since PDS selects data individually, duplicated data points may get selected in this setting. Therefore, PDS is better suited for further improving the qualities of data after de-duplication. Since data-deduplication is well-developed in recent years [1,2,3], our method focuses more on the data cleaning stage after deduplication. We have also added a discussion about this scenario in Appendix E and also provide directions for future exploration.
>
> [1] SemDeDup: Data-efficient learning at web-scale through semantic deduplication. 2023. In ICLR Workshop.
>
> [2] D4: Improving LLM Pretraining via Document De-Duplication and Diversification. 2023. In NeurIPS.
>
> [3] Deduplicating Training Data Makes Language Models Better. 2022. In ACL.

---

### Official Review · Reviewer_3Rqd · 2024-11-01

**Soundness:** 3
**Presentation:** 3
**Contribution:** 3
**Rating:** 6
**Confidence:** 3

**Summary:**

This work proposes a novel method of data selection for training machine learning models, or LLMs in particular.
It starts by theoretically formulating data selection as an optimal control problem,
where the weights of training samples are optimized in order to minimize the total area under the downstream loss curve during training with gradient descent on the weighted data.
Based on this, a practical and efficient implementation is developed:
first solve the optimal control problem on a small proxy dataset (approximately, via bi-level optimization),
and then train a data scorer (by fitting the obtained scores from the first step),
which can then be applied for training any LLM with other datasets.
Experiments show improvements over prior data selection algorithms,
in terms of test losses and scores on some commonly used benchmarks.

**Strengths:**

- The formulation of data selection as optimal control, and the theoretical connection between two seemingly unrelated fields, are interesting.

- The presentation is good overall. Ideas are clearly explained.

**Weaknesses:**

- According to the theory of this work, in particular Eq. (6), the optimal value of $\gamma$ (the weights for training samples) is an one-hot vector,
which leads to the surprising conclusion that training an LLM with a single sample is theoretically optimal.
The authors also mention this in Line 235.
My major concern is that this seems to reveal a fundamental caveat of the proposed methodology.
Also, with many approximations in deriving the practical implementation (Section 2.3),
it becomes less and less clear how much the empirical advantages of the resulting algorithm are actually relevant to the theory in Sections 2.1 and 2.2.


- Using the "scaling law" for extrapolating empirical results with small-scale models to 175B or 405B models (Table 3) doesn't feel like science.
It is not obvious why scaling laws for conventionally trained LLMs should be applicable to PDS-trained LLMs.
In addition, the goodness of fit for calculating the scaling law's coefficients is not reported, which further makes the extrapolation less convincing.


- Regarding the empirical results, the proposed method is compared with prior data selection algorithms only in Tables 1 and 2, for 160M and 470M models that are relatively small-scale.
Moreover, many scores in these results are close to random choice (e.g., 25% accuracy for MMLU in Table 2),
hence it is not clear what the minor differences between the scores of different methods really imply.

**Questions:**

- Two questions about the theory:
(a) Could you provide some intuition about why theoretically a single training sample is optimal, according to Eq. (6)?
What's missing in this theoretical analysis?
(b) In Eq. (22) where the method of Lagrange multiplier is applied, it seems that it misses a $\max_{\mu}$ term. Is this correct? If so, does the remaining analysis still hold, or need corrections as well?
I'm willing to raise my rating if these two questions can be resolved.


- Line 883, "B" and "\beta" seems like typos; should be "C" and "c" respectively?

---

> ### Author Response · Authors · 2024-11-21
> **Response to Reviewer 3Rqd (part 1/3)**
>
> We thank the reviewer for recognizing  contribution of our theoretical formulation and the clarity of our presentation. We are also encouraged to see the reviewer’s willingness to increase the score based on  our explanations of our theoretical analysis. The following is the clarification of our theoretical and empirical results:
>
> ## The intuition behind Eq. (6) (weakness #1 and question #1 (a))
>
> Eq. (6) does not imply that a single training sample is optimal. The reasons are explained below:
>
> - Eq. (6) implies that there could exist several samples having the same largest $\sum_{t=0}^T\lambda_{t+1}^\top\nabla l(x_n,\theta^*)$ values **when the model’s parameter $\theta^*$ is trained under the optimal $\gamma$** scores, i.e. $\exists S=\{n_1,n_2,\cdots,n_{|S|}\}$ such that for any $m \notin S$.
>
>   $$
>   \sum_{t=0}^T\lambda_{t+1}^\top\nabla l(x_{n_1},\theta^*) = \cdots = \sum_{t=0}^T\lambda_{t+1}^\top\nabla l(x_{n_{|S|}},\theta^*)>\sum_{t=0}^T\lambda_{t+1}^\top\nabla l(x_{m},\theta^*)
>   $$
>
>   Therefore, multiple samples in $S$ can have non-zero $\gamma^*$ scores to satisfy Eq. (6), as long as those samples not in $S$ get zero $\gamma^*$ scores. Accordingly, although "a single training sample" is a solution to Eq. (6), **there also exist many more "equally qualified" solutions (such as assigning uniform $\gamma$ values to the data points in $S$) that satisfy Eq. (6).**
>
> - As stated in lines 56-60 and 158-169 in the latest revision, the PMP condition for data selection (Theorem 2.1) is a **necessary condition** of the $\gamma$‘s optimality. This is because the two techniques we used to prove Theorem 2.1 in Appendix B, i.e., (1) time-variant PMP for discrete dynamical system (Theorem B.1) and (2) The Lagrange Multiplier method, are all necessary conditions. Therefore, the unreasonable "a single training sample" is not necessarily the optimal solution of the problem.
>
> Many empirical results, including that in our paper and previous literature in optimal control [1,2], have shown that solving this necessary condition can successfully lead to fairly good solutions, which is much like using gradient descent in deep learning even if it does not guarantee the optimality of the solution.
>
> The description in Line 235 does not mean that “training an LLM with a single sample is theoretically optimal”. Instead, it means that to solve the PMP conditions, a straightforward implementation can use the max operation to get $\gamma$ during each outer loop iteration. However, since $\theta$ has not reached its optimum yet, the sample set $S$ having the same largest $\sum_{t=0}^T\lambda_{t+1}^\top\nabla l(x_n,\theta)$ values has not emerged, making the max operation unstable in practice. Therefore, we increase $\gamma$ by a value proportional to  $\sum_{t=0}^T\lambda_{t+1}^\top\nabla l(x_n,\theta)$ in each iteration of outer loop as in Algorithm 1. This update strategy still satisfies our theory because it guarantees that the sample set having the largest $\sum_{t=0}^T\lambda_{t+1}^\top\nabla l(x_n,\theta)$ values gets the highest $\gamma$ values. It also ensures that the samples with the same $\sum_{t=0}^T\lambda_{t+1}^\top\nabla l(x_n,\theta)$ values receive the same $\gamma$ scores, which avoids the “single training sample” solution.
>
> [1] Optimal Control with Engineering Applications. 2007. Springer.
>
> [2] Optimal Control Theory with Economic Applications. 1987. Amsterdam: North-Holland.
>
> ## The Lagrange Multiplier method (question #1 (b))
>
> We omit the $\max_{\mu}$ term in Eq. (22) because this term is equivalent to the time-invariant constraint in Eq. (20). Specifically, by treating $\mu_t$ also as the time-variant control variables and applying the standard PMP conditions (Theorem B.1), the $\max_{\mu}$ term yeilds
>
> $$
> \mu_t=\arg\max_{\mu_t}\sum_{n=1}^{|D|}\mu_{n,t}(\gamma_{n,t}-\gamma_{n,0}).
> $$
>
> Since $\mu_t$ does not have any constraint, it exists and is finity ($<+\infty$) if and only if $\gamma_{n,t}=\gamma_{n,0}$ for all $n=1,2,\cdots,|D|$. The same analysis applies to all $1 \le t \le T$. In this way, we have $\mu_t=\arg\max_{\mu_t} 0$, which does not introduce additional constrains to the problem. This means the remaining analysis in our paper still holds.

---

> ### Author Response · Authors · 2024-11-21
> **Response to Reviewer 3Rqd (part 2/3)**
>
> ## The relevance between the resulting algorithm and the theory (weakness #1)
>
> Our algorithm is directly derived from Theorem 2.1.
>
> For Algorithm 1, we use Eq. (4) and Eq. (5) to compute $\theta$ and $\lambda$ in the forward and reverse inner loops. Then, we follow Eq. (6) to obtain $\gamma$  in each outer loop by updating $\gamma$ with the $\sum_{t=0}^T\lambda_{t+1}^\top\nabla l(x_n,\theta)$ values. This update strategy still aligns with Theorem 2.1 because the sample set with the largest $\sum_{t=0}^T\lambda_{t+1}^\top\nabla l(x_n,\theta)$ values still gets the highest $\gamma$. Additionally, this update ensures the samples with the same  $\sum_{t=0}^T\lambda_{t+1}^\top\nabla l(x_n,\theta)$ values are assigned the same $\gamma$, avoiding the “single training sample” solution.
>
> The effectiveness of our proposed efficient implementation is empirically verified. In Figure 6, we showed that the efficient implementation preserves most of the performance of the exact algorithm  in a simulated setting. In Figure 7, we showed that the data scorer achieves around 0.55 Spearman correlation in fitting the mapping between each data point and its $\gamma$. This is a reasonably high correlation and when we select 40% data using the trained data scorer, we find that 81.6% of the data points are correctly marked as selected/discarded in the validation set.
>
> ## The scaling law for extrapolating empirical results (weakness #2)
>
> The scaling law still applies to PDS-trained models because only the pre-training corpus is replaced with our selected data, with all other model and optimization configurations unchanged compared to conventional pre-training. Previous works have shown that the power law applies to different data sources [1,2,3,4] when all other configurations remain unchanged. Furthermore, the results in General Response show that the power law fits the observed performance data very well for both conventional pre-training and pre-training with PDS-selected data.
>
> Based on the goodness of fit of our scaling curve and the previous literature, we firmly believe that the fitted scaling curve reveals the performance trend of the PDS method in comparison to the conventional pre-training as model and data sizes scale up.
>
> [1] DeepSeek LLM Scaling Open-Source Language Models with Longtermism. 2024. arxiv pre-print.
>
> [2] Resolving Discrepancies in Compute-Optimal Scaling of Language Models. 2024. In ICML Workshop.
>
> [3] Scaling Laws for Transfer. 2021. arxiv pre-print.
>
> [4] MiniCPM: Unveiling the Potential of Small Language Models with Scalable Training Strategies. 2024. In COLM.

---

> > ### Author Response · Authors · 2024-11-21
> > **Response to Reviewer 3Rqd (part 3/3)**
> >
> > ## The empirical results (weakness #3)
> >
> > - We mainly compare our method with other data selection methods on 160M and 470M models due to the computation limits, as we would need to pre-train a language model from scratch for each method. As a supplement to our results, we test other data selection methods on 1B models and list the results below. The results show similar trend as that for 160M and 470M in our paper. We plan to complete the results on 1.7B models given more time and computation.
> >
> >
> >   |             | HS       | LAMB     | Wino.    | OBQA     | ARC-e    | ARC-c    | PIQA     | SciQ     | BoolQ    | Avg.     |
> >   | ----------- | -------- | -------- | -------- | -------- | -------- | -------- | -------- | -------- | -------- | -------- |
> >   | Convetional | 39.9     | 47.6     | 52.4     | 30.6     | 49.3     | 26.4     | 63.1     | 73.7     | 60.9     | 49.3     |
> >   | RHO-Loss    | 39.8     | 47.0     | 53.0     | 30.8     | 48.0     | 26.4     | 62.9     | 71.1     | **61.0** | 48.9     |
> >   | DSIR        | 40.8     | 47.8     | 53.0     | 31.2     | 49.8     | 26.8     | 62.7     | 76.6     | 58.0     | 49.6     |
> >   | IF-Score    | 39.4     | 47.0     | 52.6     | 28.6     | 49.4     | 26.4     | 63.5     | 74.0     | 60.5     | 49.0     |
> >   | PDS         | **42.1** | **48.8** | **54.0** | **33.4** | **51.3** | **28.0** | **64.1** | **78.5** | 58.7     | **51.0** |
> >
> > - For the MMLU test set, the better-than-random performance has not fully emerged on our model when using the “Accuracy” metric [1]. However, previous works [2,3] have shown that the performance gap between different methods can still be measured using smoother metrics in this case, like the model’s perplexity on the correct answer. Therefore, we report the perplexity of different models on the golden answers in the following table. We can see that the perplexity improves with PDS compared to the baselines. We report the accuracy in our paper because we want to show PDS helps the LM’s abilities to emerge and we have included the PPL results in Appendix H.2 as a supplement.
> >
> >
> >   |              | 160M-PPL | 470M-PPL |
> >   | ------------ | -------- | -------- |
> >   | Conventional | 44.7     | 34.8     |
> >   | RHO-Loss     | 42.1     | 33.0     |
> >   | DSIR         | 40.5     | 34.0     |
> >   | IF-Score     | 41.5     | 31.1     |
> >   | PDS          | **36.6** | **27.1** |
> >
> >   [1] Emergent Abilities of Large Language Models. 2022. In TMLR.
> >
> >   [2] Are Emergent Abilities of Large Language Models a Mirage?. 2023. In NeurIPS.
> >
> >   [3] Scaling Laws for Downstream Task Performance of Large Language Models. 2024. arxiv pre-print.
> >
> > ## Typo (question #2)
> >
> > Yes. It is a typo and we have fixed it in the revision.

---

> ### Author Response · Authors · 2024-11-23
> **We hope the reviewer find our response helpful**
>
> We highly appreciate the reviewer’s insightful feedback, which clearly helped us improve our paper.
>
> As the reviewer mentioned being *willing to raise their rating* if the two questions in Question #1 were resolved, we have provided detailed explanations on our theoretical results to address these concerns. Additionally, we have included thorough evidence to support the scaling curve extrapolation, demonstrate our method’s effectiveness on larger models, and highlight the improvements on MMLU.
>
> We sincerely hope the reviewer find our response useful and update the scores if the concerns have been resolved. We are also open to further discussion if there are further questions.

---

> ### Comment · Reviewer_3Rqd · 2024-11-24
>
> Thank you for your detailed responses.
>
> It makes sense that multiple training samples can have the same highest scores and thus non-zero weights (I played around with a few toy examples analytically and saw that this indeed happened), although the concrete values of the optimal weights cannot be found based on the proposed theory that offers necessary conditions.
>
> I think my major concerns have been resolved, so I've raised my rating as promised. It would be useful to add the clarifications in your responses for this "one-hot vector" issue, as well as the goodness of fit for scaling laws and the additional empirical results, to the revision of your manuscript.

---

### Official Review · Reviewer_snbg · 2024-11-03

**Soundness:** 3
**Presentation:** 3
**Contribution:** 4
**Rating:** 8
**Confidence:** 3

**Summary:**

The paper suggests a method for data selection based on applying optimal control theory to compute the importance of individual data points. Crucially, this importance is computed in the context of end-to-end training, to optimize the AUC of downstream performance - this is in contrast to previous works that don’t look at training trajectory. Several optimizations are used to approximate the PMP solution. Moreover, the data selection is done offline so there is just a constant cost.

**Strengths:**

- Extensive ablations (optimizing Algorithm 1, choice of downstream task) are performed to support the empirical claims and many of the choices made.
- The tackled problem is very relevant and the approach is innovative, using optimal control to account for the complete training trajectory.
- The empirical results are very promising on the small-to-medium scale experiments that they are run on. A scaling law is also computed to try to support generalization to larger models.

**Weaknesses:**

- It would be good if Appendix G3, choice of $J(\theta)$ was covered in the main body (regarding the influence of the downstream data on performance) since that is an important driver of the performance.
- Better explain Figure 1: in (b), do you have the same amount of data being used by your method and Redpajama’s to train? Do all points correspond to the same model size in (a)? If not, how do you vary these?
- It would be useful to mention at line 89 that J is not computed on the same downstream tasks that you measure one (as you mentioned at line 307) since this distinction is crucial and not made clear until line 307.
- It seems like line 100, defining $\gamma$’s makes an implicit assumption about points having a notion of independent importance, whereas one could envision cases where a point’s importance is contingent on other points being in the dataset. It would be useful to add a discussion on this - as stated, it looks like this is a fact, not a choice.
- It would be useful to add a discussion on why it makes sense to need Gumbel sampling - after all, one could expect $\gamma$’s to already have accounted for the need of diversity.

**Questions:**

- Why did you decide at line 259 to use the average of the output hidden states rather than the last one? Intuitively this gives more weight to how the datapoint starts (since the first activations at the first few positions affect both the hidden states at these positions as well as later on).
- Is it computationally feasible to constrain the $\ell_0$-norm of $\gamma$? If so, do you expect to get the same data points as you would if you took the biggest ones by the current $\gamma$?

---

> ### Author Response · Authors · 2024-11-21
> **Response to Reviewer snbg (part 1/2)**
>
> We sincerely thank the reviewer for highlighting the strengths of our work, including the extensive ablations to support our empirical claims, the innovation of our approach, and the promising empirical results. We also appreciate the reviewer's recognition of the computed scaling law as an effort to generalize to larger models.
>
> **Regarding the weakness:**
>
> ## The results of the $J(\theta)$ choice :
>
> Thanks for your constructive suggestion. Considering the strict page limit of ICLR, we chose to put the results in the Appendix. A more important reason is that the main contribution of our paper is the proposed theoretical framework and the PDS method, rather than the choice of $J(\theta)$. We would like to elaborate more on the unique key factors of PDS compared to previous methods, including its general empirical results (Table 1-3, Figure 4-5), its relationship to the theoretically optimal results (Figure 6), its computation complexity (Table 4), the effect of different incorporated learning information (Table 5), and the choice of the proxy setting (Figure 7). Furthermore, since the choice of $J(\theta)$ has been discussed in some previous studies [1,2], we do not emphasize it as one of our key findings and choose to put it in the Appendix to keep the full results within the page limit. In Section 3.1 (lines 305-306), we add a cross-reference to the results for the convenience of the readers.
>
> [1] Less: selecting influential data for targeted instruction tuning. 2024. In ICML.
>
> [2] DsDm: Model-Aware Dataset Selection with Datamodels. 2024. In ICML.
>
> ## Explanation to Figure 1:
>
> - We used the same amount of data for our method and Redpajama to train the models.
> - All the points correspond to the training of a 1.7B model in (a).
>
> ## The way $J(\theta)$ is computed:
>
> In the revision of our paper, we have clarified that $J$ is not computed on the same downstream tasks as those used in evaluation. (lines 80-82)
>
> ## The independence assumption of each example:
>
> - We have clarified the assumption that we focus on the independent importance of each example in the revision (lines 103-104) and add the following discussion to Appendix E.
>
> - Considering the dependence between each data point is a promising direction for future extension of our method. Some straightforward implementation could be improving the pair-wise difference between the examples to promote the diversity of the selected data. Specifically, we consider the pairwise similarity:
>
>   $$
>   \operatorname{sim}(x_n,x_m)=\frac{h_n^\top h_m}{||h_n||||h_m||},
>   $$
>
>   where $h_n$ and $h_m$ are the representations of each example generated by a model like RoBERTa [1]. Then, we have the following loss to minimize, which encourages the difference between examples:
>
>   $$
>   \mathcal{L}^{\text{diversity}}=\sum_{n,m}\text{sim}(x_n,x_m)\gamma_n \gamma_m = \pmb{\gamma}^\top\pmb{S} \pmb{\gamma},
>   $$
>
>   where $\pmb{S} = \left[\operatorname{sim}(x_n,x_m)\right]_{1\le n \le |\mathcal{D}|, 1\le m \le |\mathcal{D}|}$. We can add this loss to the optimization problem in Eq. (3), using a hyper-parameter $w$ to control its weight:
>
>   $$
>   \min_{\pmb{\gamma}} \sum_{t=1}^{T} J(\theta_t) + w\cdot\pmb{\gamma}^\top\pmb{S}\pmb{\gamma}.
>   $$
>
>   This problem can still be solved by the PMP conditions, and we leave the design for efficient implementation to future work.
>
>   We have included the above discussion to the Appendix E in the revision of the paper.
>
>   [1] RoBERTa: A Robustly Optimized BERT Pre-training Approach. 2019. arxiv pre-print.
>
> ## The use of Gumbel Sampling
>
> Basically, $\gamma$ is computed based on the distance between the sample gradient and the target vector $\lambda$. Therefore, if the original corpus contains too many duplicated samples, their distances to $\lambda$ will also be similar, which means that if one example gets a high $\gamma$, the others will also get high $\gamma$ and being selected. This would affect the diversity of the selected corpus. Therefore, we adopt Gumbel sampling to introduce some variance to this selection, which means even if a set of examples has similar high scores, it is still possible that not all of them will be selected. Similar techniques are also adopted in previous works [1,2].
>
> [1] MATES : Model-Aware Data Selection for Efficient Pre-training with Data Influence Models. 2024. In NeurIPS. (Section 3.1)
>
> [2] Language Models are Few-Shot Learners. 2020. In NeurIPS. (Appendix A)

---

> > ### Author Response · Authors · 2024-11-21
> > **Response to Reviewer snbg (part 2/2)**
> >
> > **Regarding the question:**
> > ## The way we compute the output hidden states
> >
> > In our pilot studies, we find that there is not much difference in the training results of the two choices. The hidden states can also be regarded as encoding the information to predict the tokens at the corresponding positions. Therefore, averaging them aggregates this information across different positions.
> >
> > ## The $l_0$-norm constraint of $\gamma$
> >
> > It is hard to implement the exact $l_0$-norm constraint because the $l_0$-norm is non-differentiable and non-convex. Computing the $l_0$-norm involves determining the exact sparsity pattern, which is equivalent to solving a combinatorial optimization problem. For high-dimensional $\gamma$, this becomes computationally prohibitive as the complexity grows exponentially with the size of the problem. In addition, the phisical significance of adding this norm constraint is unclear in our problem and is also unusual in classic optimal control problems as we do not assume the sparsity of the final $\gamma$. The phisical significance of adding norm constraints and how to implement them efficiently are good directions for future exploration.

---

> > > ### Comment · Reviewer_snbg · 2024-11-27
> > >
> > > Thank you for the response! Most of my concerns are clarified.
> > >
> > > It is still a bit unclear why you are saying the significance of $\gamma$ $\ell_0$ norm constraint would be unclear. To some extent, if you want to constrain to only using some number of datapoints, it makes sense to have exactly this many non-zero gammas. This is more so the case when you consider pairwise interactions of points where clearly the significance of a point could be altered once you take some other point out. The combo between an $\ell_0$ norm constraint and pairwise significance should also reduce the need for using Gumbel. With that said I understand and expected an $\ell_0$-norm constraint to be intractable, but thought that would be the natural target to optimize for - and if it were, I'd expect a mention of the compromise made.

---

> > > > ### Author Response · Authors · 2024-11-27
> > > > **Response to Reviewer snbg**
> > > >
> > > > We thank the reviewer for pointing out the possibility of applying the $l_0$-norm to control the sparsity of $\gamma$ so as to realize the aim of data selection. Our statement of “adding the $l_0$-norm is unclear” is under the context of optimal control theory: to our best knowledge, we do not know any prior work that introduces sparsity constraint during the control process. *It is currently unclear to us how the constraint would affect the physical property of the dynamics and resulting the optimization problem*. For example, whether it will introduce any singularity during the dynamic process or the inaccuracy from the approximated optimization procedure (given $l_0$-norm is not differentiable) would lead to any unexpected outcome.
> > > >
> > > > But we do agree with the reviewer that from machine learning perspective $l_0$-norm is a natural choice for data selection and alternative to the designed data selection method, especially when  mutual interactions among  data points  should be considered, as removing one data point would affect the contribution of the others. This could be a promising alternative to avoid the use of Gumble sampling. We have added the possible use of $l_0$-norm in Appendix E, and discussed the challenge in computation of solving the problem with the $l_0$-norm constraint.

---

### Official Review · Reviewer_KX1L · 2024-11-04

**Soundness:** 3
**Presentation:** 4
**Contribution:** 3
**Rating:** 8
**Confidence:** 3

**Summary:**

This paper proposes a data selection method (called PDS) for language model pretraining based on optimal control principles. Specifically, the PDS method aims to compute a data score for re-weighting data, such that the trained model (via a fixed procedure) achieves the best downstream test error. In practice, this is done using small-scale proxies. Additionally, a model is fitted to predict scores for general data. One important benefit of the proposed method is its offline nature, enabling an effortless plug-in to existing language model training pipelines. The effectiveness of the proposed method is verified across multiple benchmarks.

**Strengths:**

- The paper is well-written, and the experimental setup is clearly explained.
- The PDS method is well-motivated by control theory and demonstrated to be effective in practice.
- The offline nature of the proposed method makes it very easy to use in existing language model training pipeline.
- Multiple thoughtful ideas are proposed to accelerate the implementation of PDS.

**Weaknesses:**

I didn’t observe significant weaknesses in the work. If I had to mention one, it would be the lack of large-scale verification of the method's effectiveness. This limits the strength of some claims (e.g., fitting scaling laws and extrapolating results). However, this is understandable given the substantial computational resources required for such large-scale experiments.

Overall, I think the paper demonstrates the promise of the proposed method, and I am inclined to support its acceptance.

**Questions:**

Theorem 2.1 seems related to the KKT conditions of the optimization problem in equation (3). Could you briefly discuss the connection between Theorem 2.1 and the KKT conditions? Of course, this is not a weakness of the paper.

---

> ### Author Response · Authors · 2024-11-21
> **Response to Reviewer KX1L**
>
> We sincerely thank the reviewer for recognizing the good motivation and practical value of our PDS method, the clarity of our writing, and the thoughtful ideas to accelerate its implementation. We deeply appreciate the reviewer's understanding of the computational resource constraints for larger-scale experiments and the reviewer's inclination to support the acceptance of our paper.
>
> ## The large-scale experiments (the weakness)
>
> As the reviewer has mentioned, we did not conduct larger-scale experiments because of the prohibitively high computational cost since we have to train the LMs from scratch each time. Therefore, we fit the scaling curve and extrapolate the results to show the performance trend of our method compared to the baselines, as training scales up. The results in the General Response show that the scaling law curves fit our observed data well.
>
> Previous work[1] has shown that the scaling curves fitted by performance data obtained based on 40M to 2B models can provide useful guidance for large-scale training. Another work[2] uses performance data obtained under a $3\times 10^{20}$ FLOPs training budget to fit a scaling curve and accurately predicts the performance of 7B and 67B models, while our performance data are obtained under an even larger $5\times 10^{20}$ FLOPs training budget.
>
> Based on the fitting goodness of our curve and the previous literature, we firmly believe that the fitted scaling curve reveals the performance trend of the PDS method compared to conventional pre-training when model and data sizes scale up.
>
> [1] MiniCPM: Unveiling the Potential of Small Language Models with Scalable Training Strategies. 2024. In COLM.
>
> [2] DeepSeek LLM Scaling Open-Source Language Models with Longtermism. 2024. arxiv pre-print.
>
> ## Connection between Theorem 2.1 and KKT conditions (the question)
>
> The connection between Theorem 2.1 and KKT conditions is that, they are both built on the method of Lagrange multipliers. $\lambda$ is essentially the Lagrange multiplier in the conventional PMP conditions; and as the proof in Appendix B shows, we use Lagrange multiplier method to adapt the conventional PMP conditions to our data selection setting. One notable difference is that, Theorem 2.1 is a stronger result than the KKT conditions as it is applicable for the cases where the Hamiltonian function $H(\theta, \lambda, \gamma)$, as defined in Eq. (17), is not differentiable with respect to $\gamma$ [1].
>
> [1] Lawrence Craig Evans. An introduction to mathematical optimal control theory. University of
> California, 2005

---

> ### Author Response · Authors · 2024-11-25
> **We hope the reviewer find our response helpful**
>
> We sincerely thank the reviewer for the thoughtful comments and the inclination to support the acceptance of our paper.
>
> Regarding the mentioned weakness, we have provided the goodness of fit for the scaling curves in the General Response. The high correlation ($R^2$) values for both model and data fits demonstrate strong alignment, which strengthens our claims on fitting and extrapolating the scaling curves. Additionally, we reference prior literature showing that scaling curves derived from experiments of a similar scale to ours can accurately predict the performance of larger models (e.g., 7B and 67B).
>
> In response to the question, we have clarified the connection between Theorem 2.1 and the KKT conditions. Both are derived using Lagrange multipliers, but Theorem 2.1 provides a stronger result for non-differentiable dynamic systems compared to the KKT conditions.
>
> As we are approaching the end of the discussion stage, we would appreciate it if you could read our responses and update the scores if your concerns have been addressed. We are more than happy to further discuss any concerns that you find not fully addressed. Thank you.

---

> > ### Comment · Reviewer_KX1L · 2024-11-28
> > **Thank you for your response**
> >
> > Thank you for your response. I support the acceptance of this paper, as the proposed method is promising and I didn't see any significant weaknesses. I was planning to give a score of 7, but there is no such option. I will update my score to 8 to firmly support this paper!

---

### Official Review · Reviewer_tANF · 2024-11-08

**Soundness:** 3
**Presentation:** 4
**Contribution:** 3
**Rating:** 8
**Confidence:** 2

**Summary:**

The paper introduces a new data selection methodology inspired by control theory. The authors design a control problem that corresponds to training dynamics of a data-reweighted objective where the control action is the reweighting, the dynamics are gradient descent or Adam dynamics on the parameters, and the objective is minimal training error of the trajectory induced by such dynamics.  This is similar to some works in the learned optimizer literature but the optimizer parameters being optimized correspond to data selection.

On a small proxy LM, the algorithm applies a fixed-point iteration (which is shown to be projected gradient descent) in order to induce Pontryagin's Maximum Principle for this control problem and to produce a dataset of quality scores (example weights).  A model is then learned to label examples, which can be used in a data filtering process for a much larger training run.  The authors evaluate this methodology on 160M to 1.7B parameter models with some initial promising results in comparison to baseline data selection approaches.

**Strengths:**

As far as i can tell, this approach is novel, in that similar methods have not been applied to the data selection setting. This is a theoretically motivated approach that attempts to optimize the training trajectory directly.  The idea to learn a labeler to translate the ideas from leaned optimizers into a data filtering mechanism is very clever.

The paper is very well written in my opinion--the general algorithm pipeline is quite complicated, but the main technical/algorithmic ideas were well motivated and clear.

The algorithm seems to perform well against baselines, although the compute requirements are likely higher (though maybe not prohibitively so as can be seen in Table 4).

**Weaknesses:**

I am very skeptical of the scaling laws presented in this work.  There does not appear to be enough data to fit such a complicated model, and I do not think these results should be included nor trusted.

While the empirics seem to be better than baselines, there seems to be fairly marginal improvements across the board and without very careful hyperparameter sweeps on the downstream optimizer, there is reason to be skeptical of the empirics.

**Questions:**

Seems A single step of the inner loop gets most of the gains, why do you think this is?  Given this is much cheaper is this a more feasible algorithm.

Does the proxy model and the scorer models complexity need to scale as we scale up the final pretrained model we want to use this with?

Line 883, what is B and what is beta?

Typo:
Line 68: "(e.g., 125M paramaters)"

---

> ### Author Response · Authors · 2024-11-21
> **Response to Reviewer tANF**
>
> We thank the reviewer for the thoughtful comments. We are encouraged to find that the reviewer appreciates the novelty of our method and the clear presentation of our paper!
>
> ## The scaling laws (weakness #1)
>
> As described in Section H.3 in the latest revision, the model of the scaling law consists of 5 parameters, while we have 80 observed data points to fit the scaling law curves. As we explain in the following, this is considered statistically sufficient. And moreover, we did not use more data mainly because of the substantial computational resources required for larger-scale experiments.
>
> From the results in the General Response, we can clearly see that the scaling law curve fits our observed data well, in both the model and data size dimensions.
>
> We also want to point out that there are quite a few existing works fitting the scaling curves and predicting performance for large models with few much less data points. For example,
>
> - [1] fits the model performance curve with 5 points and shows good prediction in its Figure 2.
> - [2] uses 36 data points obtained from models ranging from 40M to 2B to fit a scaling curve, which provides useful guidance for large-scale training.
> - [3] selects 8 data points obtained under the training budget less than $3\times10^{20}$ FLOPs to fit a scaling curve and accurately predicts the performance of 7B and 67B models, while the maximum training computation in our experiments is larger ($5\times 10^{20}$ FLOPs), as shown in our Figure 1.
>
> Based on the goodness of fit results obtained by our experiments and practices in previous literature, we firmly believe that the fitted scaling curve reveals the trend of our PDS method's performance compared to the conventional pre-training when model and data sizes scale up.
>
> [1] GPT-4 Technical Report. 2023. arxiv pre-print.
>
> [2] MiniCPM: Unveiling the Potential of Small Language Models with Scalable Training Strategies. 2024. In COLM.
>
> [3] DeepSeek LLM Scaling Open-Source Language Models with Longtermism. 2024. arxiv pre-print.
>
> ## The empirics (weakness #2)
>
> - We achieve consistent improvement on nearly all 9 benchmarks compared to the baselines. To make the results' significance clearer: conventional pre-training has to spend 2x more  computation than PDS to bridge this gap, which can be translated as saving the training time from 7 days to 3.5 days to train a 1.7B model on 8 A100 GPUs. Therefore, our proposed method can strongly benefit the current pre-training practices.
> - Regarding the question of “hyper-parameter sweeps on the downstream optimizer”: The results reported in our paper are all zero-shot or few-shot performance of a base pre-trained model, without fine-tuning on downstream tasks. Therefore, the required results about careful hyper-parameter sweeps on the downstream optimizer are not suited for our paper. For the hyper-parameters of the model architectures, optimizers, and pre-training recipes, since we only changed the pre-training data sources of the models, we simply kept these hyper-parameters the same for a fair comparison.
>
> ## The gain of a single inner step (question #1)
>
> This is because the “target vector” $\lambda_t$ comprises two components of model information: $\nabla J(\theta_t)$ and $\eta \nabla^2L(\theta_t,\gamma)\lambda_{t+1}$, as described in Eq. (5). $\nabla J(\theta_t)$ is the zero-order information and $\eta \nabla^2L(\theta_t,\gamma)\lambda_{t+1}$ is a first-order correction when the learning rate $\eta$ is small, especially for large models. For the “single inner step” ($T=1$) case, we have $\lambda_1=\lambda_T=0$, and thus only $\lambda_0=\lambda_1+\nabla J(\theta_0)-\eta \nabla^2L(\theta_0,\gamma)\lambda_{1}=\nabla J(\theta_0)$ is computed as the “target vector” in Algorithm 1. As a result, “single inner step” can be regarded as a zero-order approximation of our solution, ignoring the first-order correction.
>
> ## The proxy and data scorer model (question #2)
>
> It is better to scale up the proxy and data scorer model because a large model may need more complex signals for data selection. However, considering the trade-off between computation and final performance, we showed that using a ~100M model to select data for a 10x larger model leads to fairly good results.
>
> ## Typos
>
> Thanks for pointing this out, and it is indeed a typo in line 883. $B$ and $\beta$ should be $C$ and $c$. We have fixed this together with the typo in line 68 in the latest revision.

---

> > ### Comment · Reviewer_tANF · 2024-11-25
> >
> > Thanks for the detailed rebuttal, especially with the goodness of fit results for the scaling law.
> >
> > I am still somewhat concerned with only 4 model sizes being used, maxing out at 2B, especially when there are reasons to think the gains of a fixed proxy model/data scorer may cap out at a certain point.
> >
> > I think perhaps my second question was poorly written; sorry about that.  What I mean is that when you change the data distribution, the optimizer hyperparams may also need to be updated and it seems like this work uses a fixed setup.  While I understand doing extra sweeps could be overly expensive, I think this is a limitation and should be addressed.  To make it clear, there is enough novelty and a sufficiently thorough evaluation that this is not grounds for rejection by any means, but I think this should be mentioned ideally.
> >
> > Overall, my opinions of the paper have not changed substantially.

---

### Public Comment · ~Hao_Di2 · 2024-11-13

Thank you for your excellent work.

I have a question regarding Algorithm 1. It appears that the vanilla trajectory $\theta_{t+1}=\theta_t - \eta \nabla L(\theta_t, \gamma)$ is treated as the solution to the motion equation defined in Eq.(4). While, if the outer loop is executed only once, and $\gamma$ is initialized as $[\dots, \frac{1}{\vert \mathcal{D}\vert}, \dots]$, Would this approximation result in a significant gap compared to the optimal trajectory?

Additionally, the results are quite interesting: in Table 1, the conventional method outperforms all baselines except the proposed method. it seems that modifications of these baselines compared to the conventional one do not lead to improved performance?

---

> ### Author Response · Authors · 2024-11-21
> **Response to Public Comments by Hao Di**
>
> Thanks for your appreciation and interest our work!
>
> ## Algorithm 1
>
> As shown in Eq. (3), $\theta_{t+1}=\theta_t-\eta \nabla L(\theta_t,\gamma)$ serves as the constraint of the trajectory of $\theta_t$, ensuring that the model is still optimized by GD when the optimal $\gamma$ is applied. Executing the outer loop once is only an implementation choice for efficiency and in practice, users can run the Algorithm 1 with multiple outer loop iterations, given sufficient computation budget. In Figure 6, we compared the $J(\theta)$ values of this one iteration implementation and the multi-iteration implementation. We can see that the efficient implementation preserves most of the performance of the exact implementation.
>
> ## Results in Table 1
>
> The performance of the baseline methods depends on the downstream datasets. In Table 2 and Table 7 in the revision, the baseline methods improve the conventional pre-training on MMLU in most cases. The baseline methods did not consistently outperform conventional pre-training for the following reasons:
>
> - RHO-Loss [1] selects instances based on the loss difference of a fully pre-trained model and a model fine-tuned on the downstream task LIMA, which is originally evaluated in the image classification task on relatively small models. It uses the final model parameters to compute the loss differences, which may be helpful for the final model training stages. However, it ignores the effect of data points on the early and middle stages of LM training.
> - DSIR [2] selects instances based on the n-gram overlap between the downstream data (LIMA) and the pre-training documents. This shallow signal works for models with relatively small sizes or continually pre-training models, as verified empirically in [2] and our results for the 160M model. However, for training larger LMs from scratch with more complex training dynamics, the shallow signal does not seem to be much helpful. Instead, it makes the LM to over-fit the common phrases in the downstream data, which can hurt the performance.
> - IF-Score [3] is first introduced to evaluate the influence of specific data points. Similar to RHO-Loss, it uses the final model parameters to compute the influence scores, ignoring the early training stages. Furthermore, its assumption $\nabla L(\theta_T)=0$ is hard to satisfy because the gradient norm does not seem to reach 0 when pre-training LMs [4].
>
> [1] Prioritized Training on Points that are Learnable, Worth Learning, and not yet Learnt. 2022. In ICML.
>
> [2] Data Selection for Language Models via Importance Resampling. 2023. In NeurIPS.
>
> [3] Understanding black-box predictions via influence functions. 2017. In ICML.
>
> [4] https://wandb.ai/ai2-llm/OLMo-1B/reports/OLMo-1B--Vmlldzo2NzY1Njk1#optimizer-metrics

---

### Author Response · Authors · 2024-11-21
**General Response to the Reviewers**

We sincerely thank all the reviewers for their thoughtful comments and constructive suggestions, which clearly helped us strengthen our paper. We are encouraged to find that the reviewers appreciate the novelty of our theoretical part and accordingly derived algorithm (Reviewer tANF, 3Rqd, NWbj), the efficient implementation (Reviewer KX1L, NWbj), extensive and solid experiments (Reviewer snbg, NWbj) with the support of the scaling law (Reviewer snbg), and clear presentation quality (Reviewer tANF, KX1L, 3Rqd). There are some shared comments regarding the goodness of fit for our reported scaling curves. We now first provide our answers to these common questions, and endeavor to provide individual responses to each reviewer.

## The Goodness of Fit of the Scaling Curves

We evaluate the goodness of fit of the scaling curves with respect to the training token size $D$ and model size $N$ respectively, by computing the correlation coefficient $R^2=1-\frac{\sum_i(y_i-\hat{y}_i)^2}{\sum_i(y_i-\overline{y})^2}$, where $y_i$ is the ground truth value and $\hat{y}_i$ is the prediction.

+ Regarding the training token size, to make the original problem a linear regression problem, we convert Eq. (53) into
$$
\log \left(L(N,D)-E-\frac{A}{N^\alpha}\right) = \log B-\beta \log D.
$$
Then, we consider $\log \left(l_i-E-\frac{A}{N^\alpha}\right)$ as the ground truth value for regression, where $l_i$ is the observed loss, and $\log B-\beta \log D_i$ as the prediction. For each $N\in[160M, 470M,1B,1.7B]$ we compute an $R^2$ respectively:

  | N    | $R^2$ (conventional) | $R^2$ (pds) |
  | ---- | ------------------ | --------- |
  | 160M | 0.990              | 0.992     |
  | 470M | 0.998              | 0.995     |
  | 1B   | 0.993              | 0.997     |
  | 1.7B | 0.996              | 0.993     |

- Similarly,  regarding the model size, we convert Eq. (53) into

  $$
  \log \left(L(N,D)-E-\frac{B}{D^\beta}\right) = \log A-\alpha \log N,
  $$

  to compute the corresponding $R^2$ that measures its goodness of fit. For simplicity, we only consider the models at the end of training, where $D=50\times 10^9$:

  | $R^2$ (conventional) | $R^2$ (pds) |
  | ------------------ | --------- |
  | 0.999              | 0.998     |

We can clearly see that $R^2$ in both cases are sufficiently high, suggesting the scaling curve fits the impact from both the data and model sizes very well.

---

### Meta-Review · Area_Chair_KB86 · 2024-12-19

**Metareview:**

The paper introduces a novel framework, PMP-based Data Selection (PDS), to enhance language model pre-training by leveraging optimal control principles. Reviewers enjoyed the paper’s theoretical foundation, particularly its use of Pontryagin's Maximum Principle (PMP), and its strong empirical results demonstrating improved data usage and scaling efficiency across various model sizes. The authors effectively addressed reviewer feedback during the rebuttal, clarifying technical details and strengthening their claims. Given its significance for large-scale data selection and LM training, I recommend acceptance.

**Additional Comments On Reviewer Discussion:**

During the discussion, reviewers asked for clarification on the technical presentation of the PMP-based Data Selection framework and its practical implementation. Concerns were raised about how the theoretical framework generalizes to different model scales and the clarity of experimental details. The authors addressed these concerns by refining explanations and providing additional insights into the empirical results. The reviewers were largely satisfied with the clarifications, recognizing the paper's contributions to efficient data selection for language model training.

---

### Decision · Program_Chairs · 2025-01-22

Accept (Oral)